# Personalized Negative Reservoir for Incremental Learning in Recommender Systems

**Antonios Valkanas**                                              *antonios.valkanas@mail.mcgill.ca*
*McGill University, Mila, ILLS*

**Yuening Wang**                                                   *yuening.wang@huawei.com*
*Huawei Noah's Ark Lab*

**Yingxue Zhang**                                                  *yingxue.zhang@huawei.com*
*Huawei Noah's Ark Lab*

**Mark Coates**                                                    *mark.coates@mcgill.ca*
*McGill University, Mila, ILLS*

**Reviewed on OpenReview:** *https://openreview.net/forum?id=jrUUk5Fskm*

## Abstract

Recommender systems have become an integral part of online platforms. Every day the volume of training data is expanding and the number of user interactions is constantly increasing. The exploration of larger and more expressive models has become a necessary pursuit to improve user experience. However, this progression carries with it an increased computational burden. In commercial settings, once a recommendation system model has been trained and deployed it typically needs to be updated frequently as new client data arrive. Cumulatively, the mounting volume of data is guaranteed to eventually make full batch retraining of the model from scratch computationally infeasible. Naively fine-tuning solely on the new data runs into the well-documented problem of catastrophic forgetting. Despite the fact that negative sampling is a crucial part of training with implicit feedback, no specialized technique exists that is tailored to the incremental learning framework. In this work, we propose a personalized negative reservoir strategy, which is used to obtain negative samples for the standard triplet loss of graph-based recommendation systems. Our technique balances alleviation of forgetting with plasticity by encouraging the model to remember stable user preferences and selectively forget when user interests change. We derive the mathematical formulation of a negative sampler to populate and update the reservoir. We integrate our design in three SOTA and commonly used incremental recommendation models. We show that these concrete realizations of our negative reservoir framework achieve state-of-the-art results for standard benchmarks using multiple top-k evaluation metrics.

## 1 Introduction

Recommender systems have become a crucial part of online services. Delivering highly relevant item recommendations not only enhances user experience but also bolsters the revenue of service providers. The advent of deep learning-based recommender systems has significantly elevated the quality of user and item representations (Covington et al., 2016; Guo et al., 2017; Cheng et al., 2016; He et al., 2023). To more accurately represent user behavior, there has been a substantial expansion in the volume of training data, accumulated from the long user-item interaction history (He et al., 2023; Xu et al., 2020). Thus, the exploration of larger and more expressive models has become a vital research direction. For example, Graph Neural Network (GNN) based recommendation methods can achieve compelling performance on recommen-

dation tasks because of their ability to model the rich relational information of the data through the message passing paradigm (van den Berg et al., 2018; Wang et al., 2019b; 2021a; Sun et al., 2019).

However, this evolution brings with it a potential increase in computational burden. An industrial-scale recommendation serving model, once integrated into an online system, usually requires regular updates to accommodate the arrival of recent client data. The constant arrival of new data inevitably leads to a point where full-batch retraining of the model from scratch becomes infeasible. One straightforward way to tackle this computational challenge is to train the backbone model in an incremental fashion, updating it only when a new data block arrives, instead of full batch retraining with older data. In industrial-level recommender systems, the new data block can arrive in a daily, hourly or even in a shorter interval (Xu et al., 2020; Wang et al., 2021b), depending on the application. Unfortunately, naively fine-tuning the model with new data leads to the well-known issue of "catastrophic forgetting" (Kirkpatrick et al., 2017), *i.e.*, the model discards information from earlier data blocks and overfits to the newly acquired data. There are two mainstream methods to alleviate the catastrophic forgetting problem: (i) experience (reservoir) replay (Prabhu et al., 2020; Ahrabian et al., 2021); and (ii) regularization-based knowledge distillation (Castro et al., 2018; Kirkpatrick et al., 2017; Xu et al., 2020; Wang et al., 2021b; 2023). Reservoir replay methods retrain on some previously observed user interactions from past data blocks while jointly training with the new data. Regularization techniques are typically formulated as a knowledge distillation problem where the model trained on the old data takes the role of the "teacher" model and the model fine-tuned on the new data is regarded as the "student" model. A knowledge distillation loss is applied to preserve the information from the teacher to the student model through model weight (Castro et al., 2018; Kirkpatrick et al., 2017), structural (Xu et al., 2020), or contrastive distillation (Wang et al., 2021b).

We note there is one important aspect of the incremental learning framework for graph based recommender systems that has received very little attention. Negative sampling plays a critical role in recommendation system training. This is because when training with implicit feedback we do not have access to explicit user dislikes so we need a mechanism to select low-user interest items for training. Sampling negative items is necessary as training on all possible positive and negative item pairs would be too costly. A good negative sampler can lead to increased convergence speed. This is key in live deployment settings where there is limited amount of time to train before a new batch of data is generated. The most common strategy for negative sampling involves uniform sampling (Rendle et al., 2009). While subsequent works improve upon the negative sampler for the static setting, there are no works dedicated to addressing negative sampling in incremental learning. *We identify two unique challenges* for designing a good negative sampler for an incremental learning framework. First, the negative reservoir must be personalized for each user and it should model a user's interests or preference shift across time blocks. This can provide a significant distribution bias in the negative sampling process. This relates to a classic trade-off in continual learning, namely achieving a correct balance between retaining knowledge learned from prior training (*stability*), and being flexible enough to adapt to new patterns (*plasticity*) in a delta-regime (Oreshkin et al., 2024) that combines the two. Second, the ranking (prediction) produced by the model from the previous time block should be exploited as a baseline for informative negative samples in the new block. We note that the focus of our work is on accounting for user interest shifts rather addressing the problem from a traditional continual learning perspective that focuses on preventing catastrophic forgetting (Wang et al., 2023). The reason for this is that, as we show in the experiments and case study, existing incremental learning techniques are too focused on *stability* and neglect model *plasticity*.

Considering the above-mentioned challenges, we design the first negative reservoir strategy tailored for the incremental learning framework. This negative reservoir contains the most effective negative samples in each incremental training block based on the user change of interest. Methodologically this translates into two hypotheses concerning the properties of a good sampling distribution for negative items. Firstly, the negative sampler should respond to user changes of interest and sample more items from item categories that the user is losing interest in over time and secondly the sampler should prefer items that are ranked highly (hard negatives) to induce larger gradients and learn faster. Our negative reservoir design is compatible with any incremental learning framework that employs negative sampling, which includes the standard incremental learning frameworks for recommender systems. In our experiments, we integrate our negative reservoir design into three recent incremental learning frameworks. Our designed negative reservoir achieves state-

of-the-art performance when incorporated in three standard incremental learning frameworks, improving GraphSAIL (Xu et al., 2020), SGCT (Wang et al., 2021b), and LWC-KD (Wang et al., 2021b) by an average of 11.2%, 8.3% and 6.4%, respectively, across six large-scale recommender system datasets. Our main contributions can be summarized as:

- This is the first work to propose a negative reservoir design tailored for incremental learning in graph-based recommender systems. The approach, for which we provide a principled mathematical derivation in Section 5.1, is compatible with existing learning frameworks that involve triplet loss.

- We demonstrate the effectiveness of our personalized negative reservoir via a thorough comparison on six diverse datasets. We strongly and consistently improve upon recent SOTA incremental learning techniques (Sections 6.3). These results are not achievable with other negative samplers (Section 6.4).

- We propose a personalized negative reservoir based on the user-specific preference change. All prior works assume static user interests when selecting negative items. In Section 6.6 we demonstrate that for users with high interest shift our model can quickly adapt and clearly improves recommendations.

## 2 Related Work

### 2.1 Incremental Learning for Recommendation Systems

Incremental learning is a training strategy that allows models to update continually as new data arrive. However, naively fine-tuning the model with new data leads to "catastrophic forgetting" (Kirkpatrick et al., 2017; Shmelkov et al., 2017; Castro et al., 2018), i.e., the model overfits to the newly acquired data and loses the ability to generalize well on data from previous blocks. There are two main research branches that aim to alleviate the catastrophic forgetting issue. The first direction is called reservoir replay or experience replay. Well known works such as iCarl (Rebuffi et al., 2017) and GDumb (Prabhu et al., 2020) construct a reservoir from the old data and replay the reservoir while training with the new data. The reservoir is usually constructed via direct optimization or heuristics. In the next paragraph we discuss methods related to incremental learning for graph-based recommendation systems. We note that there is extensive literature outside this area, for example in online learning for CTR prediction (Aharon et al., 2017) and general online or incremental learning (Valkanas et al., 2024; Waxman & Djuric, 2024) that we do not cover because it is beyond the scope of this work.

A recent incremental framework for graph-based recommender systems extended the core idea of the GDumb heuristic and proposed a reservoir sampler based on node degrees (Ahrabian et al., 2021). Another line of research focuses on regularization-based knowledge distillation (KD). The model trained using old data blocks serves as the teacher model, and the model that is fine-tuned using the new data is the student. A KD loss (Hinton et al., 2015) is applied to preserve certain properties that were learned from the historical data. In GraphSAIL (Xu et al., 2020), each node's local and global structure in the user-item bipartite graph is preserved. By contrast, in SGCT and LWC-KD (Wang et al., 2021b), a layer-wise contrastive distillation loss is applied to enable intermediate layer embeddings and structure-aware knowledge to be transferred between the teacher model and the student model. However, one important aspect of the incremental learning framework that has received very little attention is how to properly design a negative sample reservoir. This is a key omission considering the important role of negative sampling in recommendation. In this paper, we shed some light on how to design a negative reservoir tailored to the special characteristics of incremental learning in recommender systems.

### 2.2 Negative Sampling in Recommendation Systems

Since the number of non-observed interactions in a recommendation dataset is vast (often in the billions (Ying et al., 2018a)), sampling a small number of negative items is necessary for efficient learning. Random negative sampling is the default sampling strategy in Bayesian Personalized Ranking (BPR) (Rendle et al., 2009). Some more recent attempts aim to design a better heuristic negative sampling strategy to obtain more effective negative samples from non-interacting items (Rendle & Freudenthaler, 2014; Caselles-Dupré et al., 2018; Zhao et al., 2015; Ying et al., 2018a). In general, the intuition is that presenting "harder" negative

samples during training should encourage the model to learn better item and user representations. Some heuristics select negative samples based on the popularity of the item (Rendle & Freudenthaler, 2014) or the node degree (Caselles-Dupré et al., 2018). Some strive to identify hard negative samples by rejection (Zhao et al., 2015), or via personalized PageRank (Ying et al., 2018a). Other works focus on a more sophisticated model-based negative sampler. For example, DNS (Zhang et al., 2013) chooses negative samples from the top ranking list produced by the current prediction model. IRGAN (Wang et al., 2017) uses a minimax game realized by a generative adversarial network framework to produce negative sample candidates. Yu et al. (2022) address the issue of class imbalance of negatives in a static recommendation setting. While not strictly proposed for recommendation systems, Yao et al. (2022) and Chen et al. (2023) propose negative samplers for knowledge graphs that aim to sample "hard" negatives while minimizing the chance of sampling false negatives, i.e., unobserved true positives. Although these negative sampling strategies yield improvement when applied naively to incremental recommendation, they do not take the time-evolving interests of users into account and as such leave substantial room for improvement in this specific setting. Additionally, the GAN techniques often rely on reinforcement learning for the optimization. In industrial settings, the instability of GAN optimization is highly undesirable.

## 3 Problem Statement

In this section, we provide a clear definition of the incremental learning setting and our problem statement. Consider a bipartite graph $\mathcal{G}_t = (\mathcal{U}_t, \mathcal{I}_t, \mathcal{E}_t)$ with a node set $\mathcal{V}_t = \mathcal{U}_t \bigcup \mathcal{I}_t$ that consists of two types: user nodes $\mathcal{U}_t$ and item nodes $\mathcal{I}_t$. A set of edges $\mathcal{E}_t$ interconnects elements of $\mathcal{U}_t$ and $\mathcal{I}_t$; thus each edge of $\mathcal{G}_t$ encodes one user-item interaction. As is done frequently, in a recommendation setting, we only have access to implicit feedback data. Concretely, this means that for each user $u$ we have access to the set of items that a user interacts with $\mathcal{I}_u^+ = \{i : (u, i) \in \mathcal{E}\}$ but no explicit set of user dislikes $\mathcal{I}_u^-$. We consider learning in discrete intervals, and use the integer $t$ to index the $t$-th interval. This corresponds to a continuous time interval $[t\Delta T, (t+1)\Delta T)$. When we refer to the interactions at time t, indicated as $\mathcal{E}_{(t,t+1]}$, we thus mean all interactions in the interval $[t\Delta T, (t+1)\Delta T)$. The graph and its component nodes and edges are indexed by integer time $t$ as they evolve over time in a discrete fashion. The graph update rule is:

$$\mathcal{G}_{t+1} = \left(\mathcal{U}_t \cup \mathcal{U}_{(t,t+1]}, \mathcal{I}_t \cup \mathcal{I}_{(t,t+1]}, \mathcal{E}_{(t,t+1]}\right), \tag{1}$$

where $\mathcal{U}_t, \mathcal{I}_t, \mathcal{E}_t$ represent the cumulative user-item interactions up to and including time $t\Delta T$ and $\mathcal{U}_{(t,t+1]}, \mathcal{I}_{(t,t+1]}, \mathcal{E}_{(t,t+1]}$ represent the user-item interactions accrued during the time interval $[t\Delta T, (t+1)\Delta T)$. Note that in our setting we do not construct a graph using the old set of edges $\mathcal{E}_t$; we merely have the observed edges from the current time block and all the user and item nodes from previous and current time blocks. For the rest of this analysis, the user-item interactions in the interval $(0, t]$ are called "base block data" and interactions belonging to subsequent time intervals $\{(t, t+1], (t+1, t+2], \dots\}$ are referred to as "incremental block data". The goal resembles a standard recommendation task; given $\mathcal{G}_t = (\mathcal{U}_t, \mathcal{I}_t, \mathcal{E}_{(t-1,t]})$, we are expected to provide a matrix $\mathbf{R}^{|\mathcal{U}_t| \times |\mathcal{I}_t|}$ ranking all items in order of relevance to each user. The main difference with respect to a standard static recommendation task is that, due to the temporal nature of the incremental learning, our training set includes data up to block $t_T$ but we aim to predict item ranking for time $t_{T+\Delta T}$. To measure the quality of recommended items we employ the standard evaluation metrics for the "topK" recommendation task, Recall@K, Precision@K, MAP@K, NDCG@K (definitions in Appendix A).

We note the difference between our setting and the distinct problem of sequential learning. Incremental learning aims to ameliorate the computational bottleneck for training, which usually limits the training instances to the most recent time block. To better inherit knowledge from the past data and the previously trained model, a specially designed knowledge distillation or experience (reservoir) replay is usually applied. In contrast, the sequential recommendation problem focuses on designing time-sensitive encoders (Hidasi & Karatzoglou, 2018; Kang & McAuley, 2018; Fan et al., 2021) (*e.g.*, memory units, attention mechanisms) to better capture users' short and long-term preferences. Thus, incremental learning is a training strategy, whereas sequential learning focuses on specific model design for sequential data. Indeed, incremental learning training strategies can be applied to different types of backbone models that may be sequential or not. Our negative reservoir tracks shifts in user interest over time; however, we do not apply this approach to new users, as there is no historical interaction data available to observe such shifts. We consider the cold-start

problem to be distinct from our setting. We note that, since our framework assumes small and frequent updates for each data block, the proportion of new users and items within each incremental training block is generally small. The cold-start user and item representations are learned via the standard GNN backbone and improving performance for them is beyond our scope.

## 4 Preliminaries

In this section, we provide the relevant background for our methodology. We succinctly review how a modern recommender system is trained using triplet loss and how knowledge distillation is applied in the incremental learning setting to alleviate forgetting. This section also serves to introduce notation.

### 4.1 Graph Based Recommendation Systems

Consider a standard static recommendation task given a bipartite graph $\mathcal{G}$ representing interactions between users and items. The typical model uses a message passing framework implemented as a graph neural network (GNN) where initial user and item features or learnable embeddings $\mathbf{e}_u$ and $\mathbf{e}_i$ are passed through a $K$-layer GNN. The messages across layers for user node $u$ can be recursively defined as:

$$\mathbf{a}_v^{(k)} = \text{AGGREGATE}\left(\left\{\mathbf{h}_u^{(k-1)} : u \in \mathcal{N}(v)\right\}\right), \tag{2}$$

$$\mathbf{h}_v^{(k)} = \text{COMBINE}^{(k)}\left(\mathbf{h}_v^{(k-1)}, \mathbf{a}_v^{(k)}\right). \tag{3}$$

Here, $\mathbf{a}_v^{(k)}$ summarizes the information coming from node $v$'s neighborhood (denoted by $\mathcal{N}(v)$). The following step, COMBINE, combines this neighborhood representation with the previous node representation $\mathbf{h}_v^{(k-1)}$. At the input layer of the GNN, the initial user node embedding is fed directly to the network, i.e., $\mathbf{h}_u^{(0)} \coloneqq \mathbf{e}_u$. Item nodes go through identical aggregation and combination steps with $\mathbf{h}_i^{(0)} \coloneqq \mathbf{e}_i$. At the final layer of the GNN we obtain node representations $\mathbf{emb}_u = \mathbf{h}_u^K$ and $\mathbf{emb}_i = \mathbf{h}_i^K$ for the user and item nodes respectively. The exact choice of the sampling method for the aggregation function and the choice of pooling for the combination operation vary by architecture. To produce an estimate of the relevance of item $i$ to user $u$ we typically consider the dot product of the user and item embeddings $\hat{y}_{ui} = \mathbf{emb}_u \cdot \mathbf{emb}_i$.

### 4.2 Knowledge Distillation for Incremental Learning

Knowledge Distillation (KD) was originally designed to facilitate transferring the performance of a complex "teacher" model to a simpler "student" model (Hinton et al., 2015). In the setting of incremental learning for recommendation, the teacher model is the model trained on old data and the student model is trained on the most recent incremental block. The overall loss that we minimize takes the form:

$$\mathcal{L}_{BASE} = \mathcal{L}_{\text{TRIPLET}} + \lambda_{\text{KD}}\mathcal{L}_{KD}(\mathcal{M}_T, \mathcal{M}_S), \tag{4}$$

where $\mathcal{L}_{\text{TRIPLET}}$ is a standard triplet loss for recommendation such as the BPR loss (Rendle et al., 2009) (defined in the Appendix B) of the student model on the incremental data batch and $\mathcal{L}_{KD}$ represents the realization of the KD loss of teacher and student models, $\mathcal{M}_T$ and $\mathcal{M}_S$. The constant $\lambda_{\text{KD}}$ is the KD weight hyperparameter. Depending on the specific incremental learning technique, $\mathcal{L}_{KD}$ can take a different form and more components may be added to the overall loss function (Xu et al., 2020; Wang et al., 2020; 2021b). For concrete examples of realizations of KD losses in our setting see the Appendix B.

## 5 Proposed Method: GraphSANE

In this section we describe our proposed approach, the Graph **S**tructure **A**ware **NE**gative (**Graph-SANE**) Reservoir for incremental learning of graph recommender systems. Our technique works by estimating the user interest shift with respect to item clusters between time blocks. It then uses the estimated user change of preferences to bias a negative sampler to provide high quality negative samples for the triplet loss. The following subsections describe how we construct, update and sample from our proposed personalized negative

reservoir, and how the item clusters are obtained. We also present our overall training objective, which is end-to-end trainable. We note that our framework can be used by any graph neural network backbone that uses triplet loss and is compatible with existing incremental learning approaches.

On a high level, our method consists of three components. First, we introduce the negative reservoir in Section 5.1 where we define loss term $\mathcal{L}_{\text{SANE}}$. We detail how to update the contents of this reservoir for each user in Section 5.2. The user interest shift is computed based on clustering the items the user interacts with. One such end-to-end trainable clustering technique (with loss component $\mathcal{L}_{\text{KL}}$) from the literature is reviewed in Section 5.3. Finally, Section 5.4 where the incremental learning negative reservoir loss $\mathcal{L}_{\text{SANE}}$ and differentiable clustering loss $\mathcal{L}_{\text{KL}}$ is combined with the base model loss $\mathcal{L}_{\text{BASE}}$ to produce the overall objective that is optimized via backpropagation.

## 5.1  Derivation

To optimize the model parameters $\Theta$, triplet loss approaches randomly sample a small number of items with which user $u$ has no observed interactions: $\mathcal{I}_u^- \subset \bar{\mathcal{I}}_u^+$, where $\bar{\mathcal{I}}_u^+ = \mathcal{I} \backslash \mathcal{I}_u^+$ represents the complementary set of $\mathcal{I}_u^+$. Thus, $\mathcal{I}_u^-$ is a randomly selected set of items that user $u$ does not interact with (Rendle et al., 2009). Existing approaches sample each negative item once per user. However, in general, repeating some highly informative negatives multiple times (or weighting them more) can increase the speed of convergence. Furthermore, in the context of incremental learning the user interest shift can be used to drive a higher number of negatives from item categories that the user is losing interest in. The likelihood of a positive item $i$ being ranked above a negative item $j$ for user $u$ is:

$$p(\succ_u | \Theta) = \sigma(\hat{y}_{ui} - \hat{y}_{uj}). \tag{5}$$

Our proposed method proposes sampling negative items with replacement. Concretely, the proposed function $\text{SANE}_{\text{OPT}}$ with respect to which we optimize $\Theta$ may consider multiple copies $N_{u,j}$ of negative example $j$ for user $u$. This leads to $N_{u,j} \geq 0$ independent observations, each obeying the likelihood from equation 5:

$$\text{SANE}_{\text{OPT}} \coloneqq \ln p(\Theta \,|\succ_u) \propto \ln\left(p(\succ_u | \Theta)p(\Theta)\right)$$

$$= \ln\left(\prod_{(u,i)\in\mathcal{E}, j\in\mathcal{I}_u^-} \sigma(\hat{y}_{ui} - \hat{y}_{uj})^{N_{u,j}} p(\Theta)\right)$$

$$= \sum_{(u,i)\in\mathcal{E}, j\in\mathcal{I}_u^-} N_{u,j} \ln \sigma(\hat{y}_{ui} - \hat{y}_{uj}) - \lambda_\Theta \|\Theta\|^2, \tag{6}$$

where $p(\Theta)$ is an isotropic normal distribution. In practice we minimize the negative of equation 6 using backpropagation: $\mathcal{L}_{\text{SANE}} = -\text{SANE}_{\text{OPT}}$. Our loss is now defined. In the next section we demonstrate a concrete procedure to obtain $(N_{u,1}, \ldots, N_{u,|\mathcal{I}|})$.

## 5.2  Negative Reservoir for Incremental Learning

**Notation and Setup:** At time $t$, using a backbone recommendation model's user and item embeddings $\mathbf{emb}_u$ and $\mathbf{emb}_i$ we can obtain the estimated ranking of items for user $u$ by considering $\hat{y}_{ui} = \mathbf{emb}_u \cdot \mathbf{emb}_i$ for all $i \in \mathcal{I}$. Ordering the items by the scores $\hat{y}_{ui}$ ranks items for user $u$ (this yields row $u$ of matrix $\mathbf{R}$ from the problem statement).

**Determining user interest shift:** We propose tracking user interests by measuring the number of interactions of each user with item categories at every time step. For example, in the toy example depicted in Fig. 1, we see that there are $K = 3$ item categories. By counting the proportion of items that each user interacts with (in Fig. 1 (b)) we obtain histograms of user-item category interactions $\mathbf{H}_{u,t} \in \mathbb{R}^K$. These histograms are then normalized and projected to the simplex $\Delta_{K-1}$ (in Fig. 1 (c)). By tracking the trajectory of each user on the simplex we can surmise the user's interest shift: $\text{shift}(u, t-1, t) = \frac{\mathbf{H}_{u,t}}{|\mathbf{H}_{u,t}|} - \frac{\mathbf{H}_{u,t-1}}{|\mathbf{H}_{u,t-1}|}$, where $|\cdot|$ denotes the L1 norm. Note that our method is robust to the case where item categories are not given. In this case we cluster the items and interpret the item clusters as induced pseudo-categories.

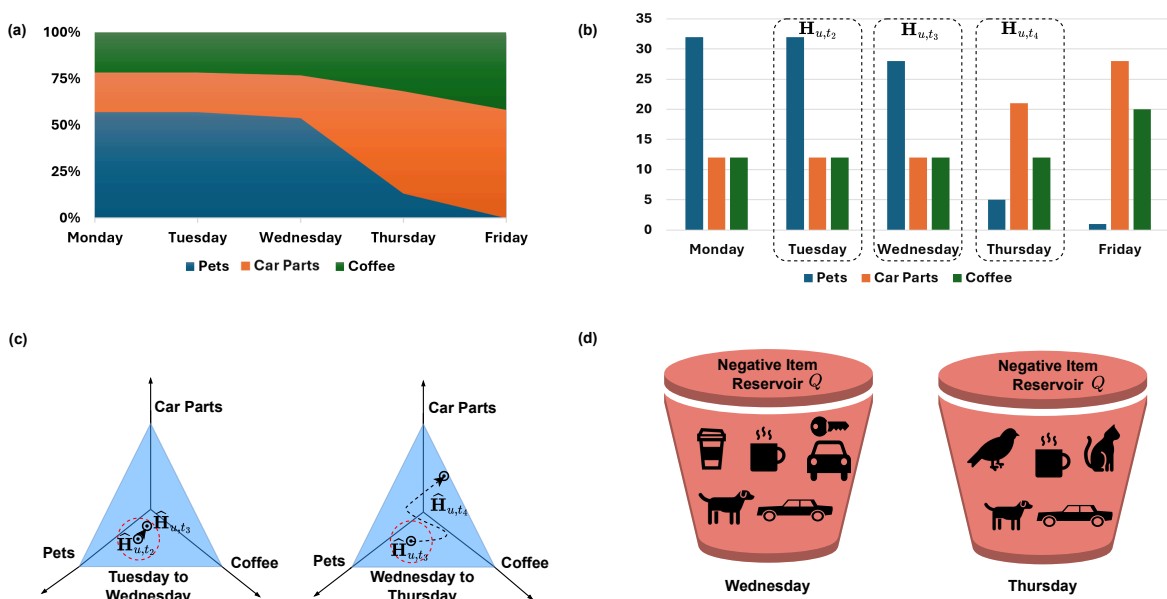

Figure 1: Toy example with daily model updates and 3 item categories: pets, car parts and coffee items. The figure shows a user's interactions with these categories over a week. (a) Cumulative user interest over time: A larger fraction of user $u$'s interactions comes from pet items at the start of the week, but his interests change over time and he interacts with more car part items by the end of the week. (b) The change of preference is reflected on the daily histograms of user-item-category interactions $\mathbf{H}_{u,t}$. (c) The histograms are normalized $\widehat{\mathbf{H}}_{u,t}$ and projected on the simplex $\Delta_2$. There we see the interest shift on the simplex representing the user's interest shift from $t = 2$, (Tuesday) to $t = 3$, (Wednesday) and on the right the user's interest shift from $t = 3$, (Wednesday) to $t = 4$, (Thursday). We see that the user exhibits a large change in interests on the second simplex (moves far away from current neighborhood in simplex denoted by a red dashed circle). (d) We propose to sample more negatives from pet category when fine tuning the model on Thursday to allow the model to quickly adjust to new user interests (in the figure the right bucket has many more pet items).

**Negative items:** Consider the top ranked $Q$ *negative* items for user $u$ (by sorting row $u$ of $\mathbf{R}$) at time $t$. This is the set of size $Q$ that contains the items that the model ranks the highest for user $u$, but which the user does not interact with. For the top $Q$ negative items per user we note the item category (or item cluster membership if categories are not available in the data) of each item. We denote $K$ as the number of item categories/clusters. This yields a count of *top negative item* interactions per item category for each user $\mathbf{c}_{u,t} = (\mathbf{c}_{u,t,1}, \ldots, \mathbf{c}_{u,t,K})$, such that $\sum_{i=1}^{K} \mathbf{c}_{u,t,i} = Q$. For example, in Fig. 1 (d), this vector would contain the counts of negative pet, car and coffee items associated with a particular user at a specific time step. We interpret the observed top negative user item-category interactions $\mathbf{c}_{u,t} \in \mathbb{N}^K = (c_{u,t,1}, \ldots, c_{u,t,K})$ as $Q$ samples of a multinomial distribution with unknown parameters $\boldsymbol{\theta}_{u,t} \in \mathbb{R}^K = (\theta_{u,t,1}, \ldots, \theta_{u,t,K})$, where $\theta_{u,t,K}$ represents the probability of sampling a negative item from category $K$ for user $u$ at time $t$.

**Our hypothesis:** A good sampling distribution for negatives should prioritize sampling negative interactions that simultaneously: (i) correspond to item categories for which the user

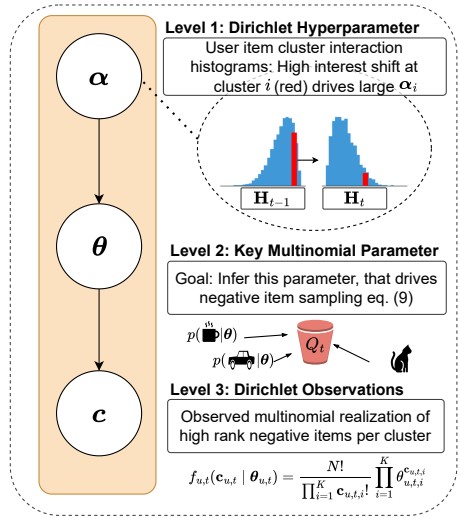

Figure 2: The hierarchical model in equation 7 & 8. From time $t - 1$ to $t$, a user loses interest in the i-th category, this increases $\boldsymbol{\alpha}_i$. Thus $\boldsymbol{\theta}_i$ has a high prior value so we sample negatives with higher probability from that item category in equation 11.

exhibits reduced interest at the present time when compared to his/her historical preferences; and (ii) are ranked highly (hard negatives)

**Hierarchical Model:** We model the prior distribution of $\boldsymbol{\theta}_{u,t}$ as a Dirichlet $\boldsymbol{\theta}_{u,t} \sim \mathbf{Dir}(\boldsymbol{\alpha}_{u,t})$. Our framework obtains a posterior distribution over $\boldsymbol{\theta}_{u,t}$ by fusing information from our prior model with the observed distribution of negative items across categories. The parameters of the prior $\boldsymbol{\alpha}_{u,t}$ are derived from the histogram of user item-category interaction counts $\mathbf{H}_{u,t}$ and $\mathbf{H}_{u,t-1}$. $\mathbf{H}_{u,t} \in \mathbb{N}^K$ is a histogram that summarizes the number of items that user $u$ interacted with from each of the $K$ categories, i.e., $\mathbf{H}_{u,t}[k] = n$ denotes that at time $t$ user $u$ interacted with $n$ items from category $k$. Recall that $Q$ is the number of negative items in each user's negative reservoir. We then define the following hierarchical model (shown in Fig. 2):

$$\text{LEVEL 1: } \boldsymbol{\alpha}_{u,t} = \lambda Q \, \text{softmax}\left( -\left[ \frac{\mathbf{H}_{u,t}}{|\mathbf{H}_{u,t}|} - \frac{\mathbf{H}_{u,t-1}}{|\mathbf{H}_{u,t-1}|} \right] \right), \tag{7}$$

where $|\cdot|$ denotes the L1 norm and $\boldsymbol{\alpha}_{u,t} \in \mathbb{R}^K$. Our choice for prior in equation 7 reflects the fact that if $\left[ \frac{\mathbf{H}_{u,t}}{|\mathbf{H}_{u,t}|} - \frac{\mathbf{H}_{u,t-1}}{|\mathbf{H}_{u,t-1}|} \right]_k < 0$, then a user is interacting with fewer items belonging to category $k$, and hence may be losing interest in the item category with index $k$, so we want to sample more negative items from it. Conversely, if the difference is positive, this indicates an increase in user interest in items from the category $k$ so the softmax function will decrease the probability of sampling a negative item from this category. The next levels are a Multinomial-Dirichlet conjugate pair:

$$\text{LEVEL 2: } \quad p(\boldsymbol{\theta}_{u,t}) = \mathbf{Dir}(\boldsymbol{\alpha}_{u,t}) = \frac{\Gamma(\alpha_{u,t,0})}{\Gamma(\alpha_{u,t,1}) \dots \Gamma(\alpha_{u,t,K})} \prod_{k=1}^{K} \theta_{u,t,k}^{a_{u,t,k}-1},$$

$$\text{LEVEL 3: } \quad f_{u,t}(\mathbf{c}_{u,t} \mid \boldsymbol{\theta}_{u,t}) = \frac{N!}{\prod_{i=1}^{K} \mathbf{c}_{u,t,i}!} \prod_{i=1}^{K} \theta_{u,t,i}^{\mathbf{c}_{u,t,i}}. \tag{8}$$

$\Gamma(.)$ is the Gamma function, $\sum_i \boldsymbol{\theta}_{u,t,i} = 1$, $\alpha_{u,t,0} = \sum_{i=1}^{K} \alpha_{u,t,i}$, $\mathbf{c}_{u,t,i}$ is the $i$-th element of $\mathbf{c}_{\mathbf{u},\mathbf{t}}$ and $\lambda \in \mathbb{R}_+$ is a positive valued hyperparameter. The posterior $p(\boldsymbol{\theta}_{u,t} \mid \mathbf{c}_{u,t}, \boldsymbol{\alpha}_{u,t})$ is:

$$p(\boldsymbol{\theta}_{u,t} \mid \mathbf{c}_{u,t}, \boldsymbol{\alpha}_{u,t}) \propto p(\boldsymbol{\theta}_{u,t}) f_{u,t}(\mathbf{c}_{u,t} \mid \theta_{u,t}) = \frac{\Gamma(\alpha_0)}{\Gamma(\alpha_1) \dots \Gamma(\alpha_K)} \prod_{k=1}^{K} \theta_k^{a_k-1} \frac{N!}{\prod_{i=1}^{K} c_i!} \prod_{i=1}^{K} \theta_i^{c_i} = \mathbf{Dir}(\boldsymbol{\alpha}_{u,t} + \mathbf{c}_{u,t})$$

$$= \mathbf{Dir}\left( \lambda Q \text{softmax}\left( \frac{\mathbf{H}_{u,t}}{|\mathbf{H}_{u,t}|} - \frac{\mathbf{H}_{u,t-1}}{|\mathbf{H}_{u,t-1}|} \right) + \mathbf{c}_{u,t} \right) \tag{9}$$

We can then estimate $\boldsymbol{\theta}_{u,t}$ as the mean of the posterior:

$$\hat{\boldsymbol{\theta}}_{u,t,i} = \mathbb{E}[\theta_{u,t,i} \mid \mathbf{c}_{u,t,i}] = \frac{\lambda Q \text{softmax}\left( \frac{\mathbf{H}_{u,t,i}}{|\mathbf{H}_{u,t,i}|} - \frac{\mathbf{H}_{u,t-1,i}}{|\mathbf{H}_{u,t-1,i}|} \right) + \mathbf{c}_{u,t,i}}{\sum_i \lambda Q \text{softmax}\left( \frac{\mathbf{H}_{u,t,i}}{|\mathbf{H}_{u,t,i}|} - \frac{\mathbf{H}_{u,t-1,i}}{|\mathbf{H}_{u,t-1,i}|} \right) + \mathbf{c}_{u,t,i}} \tag{10}$$

**Sampler for Reservoir:** Denote $\hat{\boldsymbol{\theta}}_{u,t,C}$ as the $C$-th entry of $\hat{\boldsymbol{\theta}}_{u,t}$ ($C \in \{1, \dots, K\}$ is the item category that $i$ belongs to). To sample negative items from the reservoir, we define the sampler for negative item $n_i \in \mathcal{I}$:

$$p(n_j = j, \hat{\boldsymbol{\theta}}_{u,t}) := \frac{\mathbb{I}_{[j \in \mathcal{Q}]}(j) \hat{\boldsymbol{\theta}}_{u,t,g(j)}}{\sum_i \mathbb{I}_{[i \in \mathcal{Q}]}(i) \hat{\boldsymbol{\theta}}_{u,t,g(i)}}, \tag{11}$$

where and $\mathbb{I}_{[i \in \mathcal{Q}]}(i)$ is an indicator function that is one when item $i$ belongs to the set $\mathcal{Q}$ of top negative items for user $u$ at time $t$ and $g(i)$ is a function that returns the category index of item index $i$. Connecting this to the derivation in Section 5.1, vector $(N_{u,1}, \dots, N_{u,|\mathcal{I}|})$ from equation 6 is a draw from the multinomial with parameters defined in equation 11.

### 5.3 Clustering

Recall that we often do not have pre-specified categories or item attributes that clearly identify suitable item categories. We overcome this problem by defining item pseudo-categories based on item clusters. The deep structural clustering method we use is adapted from two recent works by Bo *et al.* (Bo et al., 2020) and Wang *et al.* (Wang et al., 2019a). We note that the clustering algorithm we utilize is not a research contribution but simply an adopted method. We select this method because the clusters are learned using both node attributes and the graph adjacency, in an end-to-end trainable fashion.

For item $i$, we use the kernel of the Student's t-distribution as a similarity measure between the item embedding at time $t$, $\mathbf{h}_i^t$, and the $k$-th cluster centroid $\boldsymbol{\mu}_k^t$:

$$q_{i,k}^t = \frac{(1 + ||\boldsymbol{h}_i^t - \boldsymbol{\mu}_k^t||^2/\nu)^{-\frac{\nu+1}{2}}}{\sum_{k' \in K}(1 + ||\boldsymbol{h}_i^t - \boldsymbol{\mu}_{k'}^t||^2/\nu)^{-\frac{\nu+1}{2}}} \,, \tag{12}$$

where $K$ is the total number of item clusters and $\nu$ represents the degrees of freedom of the distribution. We consider $q_{i,k}^t$ to be the probability of assigning item $i$ to cluster $k$. Then, $Q_i^t = [q_{i,1}^t, \ldots, q_{i,K}^t]$ is a discrete probability mass function that summarizes the probability of item $i$ belonging to each cluster at time $t$. The assignment of all items can be described by $Q^t = [q_{i,k}^t]$. An issue is that $Q^t$ can sometimes provide diffuse cluster assignments (close to uniform). This is why Bo et al. (2020) obtain confident cluster assignments, *i.e.*, more mass concentration to a single cluster per item through a transformation:

$$p_{i,k}^t = \frac{(q_{i,k}^t)^2/\sum_i q_{i,k}^t}{\sum_{k'}(q_{i,k'}^t)^2/\sum_i q_{i,k'}^t}. \tag{13}$$

Then $P^t$ consists of the elements of $Q^t$ after being transformed by a square and normalization operation. By minimizing the Kullback–Leibler (KL) divergence we encourage more concentrated assignment to clusters.

$$\mathcal{L}_{\mathrm{KL}} = D_{\mathrm{KL}}(\mathrm{P}^t \,||\, \mathrm{Q}^t) = \sum_i \sum_m p_{i,j}^t \log \frac{p_{i,j}^t}{q_{i,j}^t}. \tag{14}$$

In practice we first initialize $\boldsymbol{\mu}_1, \boldsymbol{\mu}_2, \ldots, \boldsymbol{\mu}_K$ via K-means and then optimize equation 14 in subsequent training iterations so that the centroids are updated via back-propagation based on the gradients of $\mathcal{L}_{\mathrm{KL}}$. We then model the probability of item $i$ belonging to cluster $k$ using $p_{i,k}^t$ by applying a softmax function with temperature $\tau \in \mathbb{R}_+$:

$$\mathbf{c}_{i,k}^t = \frac{\exp(\mathbf{p}_{i,k}^t)/\tau}{\sum_{\hat{k}}\exp(\mathbf{p}_{i,\hat{k}}^t)/\tau}. \tag{15}$$

### 5.4 Overall Framework

In this section, we present the overall objective function, and provide an algorithm that summarizes our proposed incremental training framework. We emphasize that our proposed framework facilitates end-to-end training with any GNN backbone and any incremental learning framework. This includes both knowledge distillation and reservoir replay techniques.

While training using hard negative items can help improve gradient magnitudes and speed up convergence, it can cause instability (Ying et al., 2018a). To address this we use two sources of negative samples: (i) hard negatives from our proposed reservoir; and (ii) negatives selected randomly according to a uniform distribution. By including randomly selected negatives, we introduce diversity as well as improving stability by moderating the effects of large gradients obtained from hard negatives during the initial epochs (Ying et al., 2018a). The proposed objective is:

$$\mathcal{L}_{\mathrm{TOTAL}} = \underbrace{\mathcal{L}_{\mathrm{BASE}}}_{\mathcal{L}_{\mathrm{TRIPLET}}, \mathcal{L}_{\mathrm{KD}}} + \underbrace{\mathcal{L}_{\mathrm{SANE}}}_{\text{Neg. Reservoir}} + \beta \underbrace{\mathcal{L}_{\mathrm{KL}}}_{\text{Cluster}}. \tag{16}$$

Here $\mathcal{L}_{\mathrm{BASE}}$ consists of the base incremental learning model loss components, *i.e.*, the knowledge distillation loss $\mathcal{L}_{\mathrm{KD}}$ and a triplet loss $\mathcal{L}_{\mathrm{TRIPLET}}$, such as BPR loss, of the randomly sampled negatives. $\mathcal{L}_{\mathrm{SANE}}$ is our

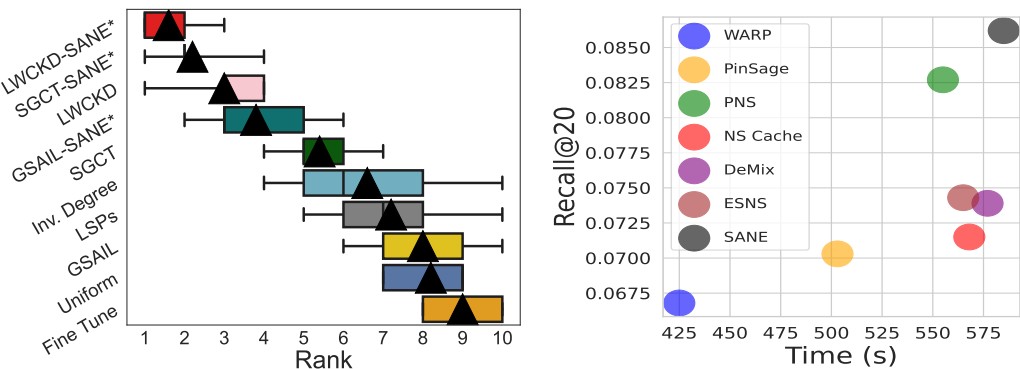

Figure 3: **Left**: Boxplot of ranks of the algorithms across the 6 datasets. The medians and means of the ranks are shown by the vertical lines and the black triangles respectively; whiskers extend to the minimum and maximum ranks. Stars "*" represent methods that integrate our proposed negative reservoir. We abbreviate GraphSail to "GSAIL". **Right**: Comparison of recall performance and training time required per incremental block for various negative samplers of Yelp discussed in Table 3. Our method (SANE) outperforms with time complexity in line with the baselines. Note that full batch retraining takes more than an hour.

proposed loss from the hard negative reservoir, and $\mathcal{L}_{\mathrm{KL}}$ is the KL loss component for the clustering. Note that both $\mathcal{L}_{\mathrm{BASE}}$ and $\mathcal{L}_{\mathrm{SANE}}$ are triplet loss based so they operate on the same scale. However, we require weighing the clustering loss term with hyperparameter $\beta$ to balance the contributions of the loss components. That is because the clustering loss operates on a completely different scale so proprely tuning $\beta$ is necessary to prevent clustering from being ignored during the optimization or dompletely dominating the objective. See Appendix K for hyperparameter tuning details. The training process with our method is detailed in Algorithm 2.

## 6 Experiments

This section empirically evaluates the proposed method **GraphSANE**. Our discussion is centered on the following research questions (RQ). Experiment code and method implementation are available online[1]:

- **RQ1** How does our method compare to standard SOTA incremental learning methods?

- **RQ2** Is our sampler better than generic negative samplers in the incremental learning setting?

- **RQ3** How does the time complexity of our sampler compare to other samplers? Is our approach scalable?

- **RQ4** Where are the performance gains coming from? For which users do we improve recommendation?

### 6.1 Datasets

We empirically evaluate our proposed method on six mainstream recommender system datasets: Gowalla, Yelp, Taobao-14, Taobao-15, Netflix and MovieLens10M. These datasets vary significantly in the total number of interactions, sparsity, average item and user node degrees, as well as the time span they cover. Detailed dataset statistics are provided in Appendix C Tab. 4. In this work we adopt the data pre-processing approach from the literature. Our data setup is identical to the baselines GraphSAIL (Xu et al., 2020), Inverse Degree (Ahrabian et al., 2021), SGCT and LWC-KD (Wang et al., 2021a). To simulate an incremental learning setting, the datasets are separated to 60% base blocks and four incremental blocks each with 10% of the remaining data in chronological order. For additional information on how the blocks are constructed see Appendix J. Fig. 5 depicts the data split per block. For the metrics, we record the performance for each incremental block and report the average.

---

[1]Link to code repository: `https://github.com/AntonValk/GraphSANE`

Table 1: Comparison (Recall@20) of baselines and 3 recent knowledge distillation algorithms with our SANE reservoir. Our method is indicated by a star (*). Best performers are in bold. Blue cell colors indicate stat. significant improvement of a particular method when our reservoir is introduced ($p < 0.05$ on Wilcoxon test).

| Methods | Gowalla-20 Average Rec@20 | Δ% | Taobao2014 Average Rec@20 | Δ% | Taobao2015 Average Rec@20 | Δ% | Yelp Average Rec@20 | Δ% | Netflix Average Rec@20 | Δ% | MovieLens10M Average Rec@20 | Δ% |
|---|---|---|---|---|---|---|---|---|---|---|---|---|
| Fine Tune | 0.1705 | 0.0 | 0.0153 | 0.0 | 0.0950 | 0.0 | 0.0661 | 0.0 | 0.1036 | 0.0 | 0.0918 | 0.0 |
| LSP_s | 0.1783 | 4.6 | 0.0152 | -0.7 | 0.0968 | 1.9 | 0.0676 | 2.3 | 0.1128 | 8.7 | 0.0925 | 0.8 |
| Uniform | 0.1728 | 1.3 | 0.0157 | 2.6 | 0.0982 | 3.4 | 0.0654 | -1.1 | 0.0957 | -7.6 | 0.0885 | -3.6 |
| Inv_degree | 0.1738 | 1.9 | 0.0175 | 14.6 | 0.0989 | 4.2 | 0.0677 | 2.4 | 0.0957 | -7.6 | 0.0915 | -0.3 |
| GraphSAIL | 0.1849 | 8.4 | 0.0155 | 1.3 | 0.0963 | 1.4 | 0.0639 | -3.3 | 0.1051 | 1.4 | 0.0890 | -3.1 |
| GraphSAIL-SANE* | 0.1925 | 12.9 | 0.0171 | 11.8 | 0.1107 | 16.5 | 0.0857 | 29.7 | 0.1086 | 4.8 | 0.0921 | 0.3 |
| SGCT | 0.1870 | 9.7 | 0.0160 | 1.7 | 0.0999 | 5.2 | 0.0668 | 1.06 | 0.1135 | 9.6 | 0.0919 | 0.1 |
| SGCT-SANE* | 0.1897 | 11.3 | 0.0178 | 16.3 | **0.1128** | **18.7** | 0.0862 | 30.4 | 0.1155 | 11.5 | 0.0946 | 3.1 |
| LWC-KD | **0.1977** | **15.9** | 0.0176 | 15.3 | 0.1030 | 8.4 | 0.0679 | 2.7 | 0.1142 | 10.2 | 0.0928 | 1.1 |
| LWC-KD-SANE* | 0.1881 | 10.3 | **0.0188** | **22.9** | 0.1114 | 17.2 | **0.0898** | **35.9** | **0.1182** | **14.1** | **0.0964** | **5.0** |

## 6.2 Baselines

Our base model recommendation system is MGCCF (Sun et al., 2019), a commonly used architecture in the incremental recommendation setting (Xu et al., 2020; Wang et al., 2021b) that is specifically designed to handle bipartite graphs. All the incremental learning algorithms are built on top of this backbone model. In our first set of experiments summarized in Table 1 (even finer-grained results per incremental block are available in Appendix Table 12), we evaluate our model in comparison to multiple baselines, including the current SOTA graph incremental learning approaches. **Fine Tune**: Fine-tune naively trains on new incremental data of each time block to update the model that was trained using the previous time block's data. It is prone to "catastrophic forgetting". **LSP_s** (Yang et al., 2020): LSP is a knowledge distillation technique tailored to Graph Convolution Network (GCN) models. **Uniform**: This method randomly samples and replays a subset of old data along with the incremental data to alleviate forgetting. **Inv_degree** (Ahrabian et al., 2021): Inv_degree is a state-of-art reservoir replay method. The reservoir is constructed from historical user-item interactions. The inclusion probability of an interaction is proportional to the inverse degree of the user. **SOTA Graph Rec. Sys. Incremental Learning methods**: GraphSail (Xu et al., 2020), SGCT (Wang et al., 2021b) and LWC-KD (Wang et al., 2021b) are state-of-the-art knowledge distillation techniques. We integrate our design into these models and compare.

In the second set of experiments, we investigate whether the proposed SANE reservoir design is effective compared to alternatives. We investigate how several prominent existing negative sampling strategies perform in incremental learning. The experiment demonstrates that the specialized sampler we propose is better than existing generic negative samplers. The methods we compare to are **WARP** (Weston et al., 2010), **PNS** (Rendle & Freudenthaler, 2014), **PinSage (sampler)** (Ying et al., 2018a), **NS Cache** (Zhang et al., 2019), **ESNS** (Yao et al., 2022), **DeMix** (Chen et al., 2023). For more information on the selected negative samplers see Appendix D.

**Hyperparameters and Reproducibility.** We make our experimental code available. We use 2 GNN layers in the MGCCF model with an embedding dimension of 128 and update the reservoir every two epochs. For a full set of hyperparameters please consult Appendix H. A detailed description on how to deploy the model and tune the hyperparameters in practical settings is in Appendix K.

## 6.3 Comparison to SOTA Incremental Learning (RQ1)

Our first set of experiments compares our method to the standard incremental learning baselines on six mainstream datasets. We use the exact same base MGCCF (Sun et al., 2019) backbone model instance trained on the base block for all incremental learning methods. We integrate our negative reservoir design SANE in three SOTA and commonly used incremental recommendation models, including GraphSAIL (Xu et al., 2020), SGCT (Wang et al., 2021b), and LWC-KD (Wang et al., 2021b). The experiment convincingly demonstrates the value of considering a negative reservoir in conjunction with standard incremental learning approaches.

We show in Table 1 that our method strongly outperforms for almost all datasets, achieving top performance in all but the smallest dataset (Gowalla-20). This is not unexpected since the smallest dataset, Gowalla, has only $\sim$5000 items. With an average user degree of $\sim$50, and assuming we select 10 negative samples per observed interaction for the optimization, even randomly sampling the negatives without replacement yields a negative sample size that is approximate $50 \times 10/5000 = 10\%$ of the total item population per epoch. Training for 10-20 epochs virtually guarantees that the pool of all potential negatives is exhausted, thus reducing the effectiveness of any non-trivial negative sampler relative to brute force sampling of all the negatives. On moderate and large datasets our method is the top performer by a convincing margin, often offering more than 10% compared to the top-performing baselines that do not use our negative reservoir. Overall, our designed negative reservoir can boost the average performance of three SOTA incremental learning frameworks, GraphSAIL, SGCT, and LWC-KD by 11.2%, 8.3% and 6.4%, respectively, across six datasets. Furthermore, as summarized in Fig. 3, LWC-KD-SANE is top-performing method overall.

Table 2: Average ranks of models for key metrics across Gowalla, Yelp, Taobao14, Taobao15. Lower rank is better ($\downarrow$). Bold indicates improvement over base model.

|  | Recall@20 Rank ($\downarrow$) | NDCG@20 Rank ($\downarrow$) | Precision@20 Rank ($\downarrow$) | MAP@20 Rank ($\downarrow$) |
|---|---|---|---|---|
| GraphSail | 6.00 | 4.75 | 5.00 | 5.00 |
| GraphSail+SANE | **3.00** | **3.75** | **3.25** | **3.50** |
| SGCT | 4.75 | 4.75 | 4.50 | 4.50 |
| SCGT+SANE | **2.25** | **2.75** | **3.25** | **2.00** |
| LWCKD | 3.25 | 3.25 | 3.75 | 3.25 |
| LWCKD+SANE | **1.50** | **2.00** | **1.50** | **2.50** |

**More Metrics: Precision, MAP, NDCG** In the previous subsection and Table 1 we only presented Recall@20 metrics. Here, we present a summary of the results of the same experimental setup as in Table 1 and Fig. 3 but also include results for Precision@k, mean average precision (MAP@k) and normalized discounted cumulative gain (NDCG@k). We show results that depict the overall rank of the methods, averaged across the datasets for $k = 20$. We ran experiments for $k \in \{5, 10, 15, 20\}$ and obtained very similar results to Table 2 in all cases. Our results span 5 datasets, 4 distinct $k$ values for 4 different metrics and pairwise comparison of 3 incremental models (using our reservoir *versus* not using it).

Table 3: Comparison of mainstream negative samplers with our proposed reservoir in terms of Recall@20. Inc 1, 2, 3 columns indicate performance for each incremental data block. Last column compares average performance across blocks. Our proposed method the average recall. Blue color cells indicate statistically significant improvement.

| Dataset | Methods | Inc 1 | Inc 2 | Inc 3 | Avg. |
|---|---|---|---|---|---|
| Yelp | SGCT+[Warp] | 0.0740 | 0.0656 | 0.0608 | 0.0668 |
|  | SGCT+[PinSage] | 0.0794 | 0.0663 | 0.0651 | 0.0703 |
|  | SGCT+[PNS] | 0.0933 | 0.0798 | 0.0748 | 0.0827 |
|  | SGCT+[NS Cache] | 0.0794 | 0.0681 | 0.0670 | 0.0715 |
|  | SGCT+[DeMix] | 0.0783 | 0.0740 | 0.0694 | 0.0739 |
|  | SGCT+[ESNS] | 0.0786 | 0.0754 | 0.0690 | 0.0743 |
|  | SGCT+[SANE] (ours) | 0.0966 | 0.0877 | 0.0744 | **0.0862** |
| Taobao14 | SGCT+[Warp] | 0.0240 | 0.0092 | 0.0148 | 0.0160 |
|  | SGCT+[PinSage] | 0.0241 | 0.0099 | 0.0151 | 0.0164 |
|  | SGCT+[PNS] | 0.0220 | 0.0114 | 0.0113 | 0.0149 |
|  | SGCT+[NS Cache] | 0.0237 | 0.0124 | 0.0121 | 0.0165 |
|  | SGCT+[DeMix] | 0.0196 | 0.0097 | 0.0113 | 0.0135 |
|  | SGCT+[ESNS] | 0.0218 | 0.0113 | 0.0128 | 0.0153 |
|  | SGCT+[SANE] (ours) | 0.0224 | 0.0136 | 0.0173 | **0.0178** |

This yields $6 \times 4 \times 4 \times 3 = 288$ distinct comparisons. **Our method improves upon the baseline in 226/288 (78.5%)** of the cases, and the **improvement is larger than 5%** in 193/288 (67.0%) of the cases. We note that the majority of the cases where our model does not improve upon the base model is in Gowalla-20, which as explained earlier is a pathological case. If we exclude Gowalla-20, **our model improves the base models $\sim$93% of the time.**

## 6.4 Comparison to Generic Neg. Samplers (RQ2)

Our second experiment focuses on evaluating the proposed negative sampler for the reservoir. The goal of this experiment is to demonstrate the effectiveness of our negative sampler in the context of incremental learning. For the baselines, we replace $\mathcal{L}_{\text{SANE}}$ with a standard triplet loss, i.e., BPR, and draw the negatives according to the baseline negative sampler algorithms. We select one incremental learning framework, SGCT, replicate

the same incremental learning setup for Yelp and Taobao2014 and vary the choice of negative samplers. As shown in Table 3, our method offers a consistent improvement. We conjecture that the performance gain arises because our sampler is the only one that is designed explicitly for the incremental setting, rather than being designed for a static setting and then adapted (this is investigated in the case study in a subsequent section). We compare the performance and the time to train one incremental block of Yelp for different samplers in Fig. 3. Our approach takes approximately the same amount of time as PNS, NS Caching, DeMix and ESNS and 20% more time than vanilla WARP.

## 6.5 Time Complexity & Efficient Scaling (RQ3)

For training complexity the main cost comes from ranking the items. Our approach exploits the fact that in a real world setting recommender systems that are trained incrementally start with a rank of the items from the previous data block. We only need to rank the items 2-3 times as the number of training epochs until model convergence is usually very low (between 5-15) for all of our datasets, even though they vary in size dramatically. Therefore, the computational cost of ranking the items a handful of times is not unreasonable. Inference time is not affected by our algorithm as we do not cluster or sample negatives during evaluation. BPR (Rendle et al., 2009) and WARP (Zhao et al., 2015) select negatives at random so they are the most efficient. The PinSage sampler (Ying et al., 2018b) requires running a personalized PageRank algorithm to rank the items. In settings where the incremental block data do not radically change the graph topology, the PageRank iterations converge quickly if the previous block's PageRank vector is used to initialize the iterative PageRank algorithm. Score and rank based models require computing dot products between user and item embeddings, which costs $O(|\mathcal{U}||\mathcal{I}|)$. These are typically the most expensive models, but they often provide "hard" negatives that can significantly improve training convergence. Such methods include our method as well as (Zhang et al., 2019; Rendle & Freudenthaler, 2014; Zhang et al., 2013). In general, ranking all the items incurs a computational cost of $O(|\mathcal{U}||\mathcal{I}|\log(|\mathcal{I}|))$. We note that we only rank the top $|\mathcal{Q}|$ items, which reduces the complexity to $O(|\mathcal{U}||\mathcal{I}|)$ on average when using QuickSelect (Hoare, 1961).

## 6.6 Case Study: GraphSANE Improves Recommendation for Dynamic Users (RQ4)

We conduct a case study on Taobao14 on the users with high interest shift to investigate their negative items drawn from the various samplers. Since our approach aims to model user interest shift, we expect our model to (i) draw more negatives from old positive items, thus validating our hypothesis that explicitly targeting user loss of interest in items for negative samples is sensible; and (ii) outperform in the recall metric, since our algorithm focuses on this subset of users. As we can see in Fig. 4, our algorithm increases the proportion of negative samples from old positives by sevenfold and offers clear improvement of about 20% in Recall@20. This compares to an improvement of 15% for the remaining 85% of users with lower interest shift. Note that this shows that

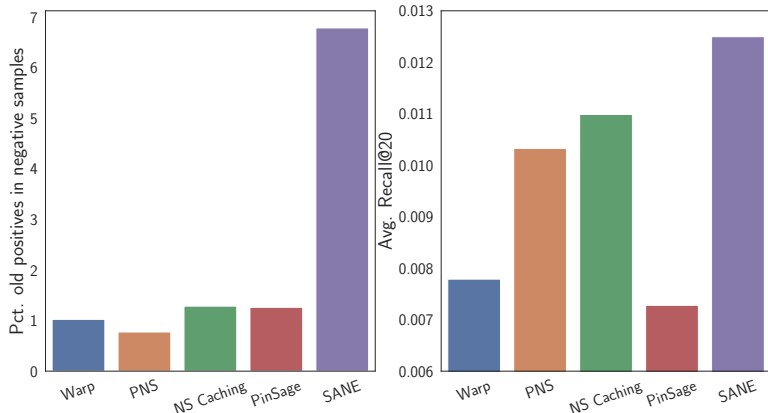

Figure 4: Case Study: Among the 15% of users with the highest interest shift on Taobao14 we observe that Graph-SANE increases the amount of old positives sampled as current negatives by 7 times. Our sampler strongly improves Recall@20.

our approach is particularly effective for high shift users relative to the general population. This further validates our hypothesis about the effectiveness of tracking user interest shift when drawing negative samples. Details for the case study are available in Appendix F.

## 7 Conclusion

This work proposes a novel incremental learning technique for recommendation systems to sample the negatives in the triplet loss. The method is not applicable to the case where users or items are not represented by their identity, but rather only by their features (i.e. no user / item nodes at all). Our approach is easy to implement on top of any graph-based recommendation system backbone such as PinSage(Ying et al., 2018a), MGCCF (Sun et al., 2019), or LightGCN (He et al., 2020a), and can be easily combined with base incremental learning methods. When used in conjunction with standard knowledge distillation approaches, our method demonstrates a very strong improvement over the current state-of-the-art models.

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

# Personalized Negative Reservoir for Incremental Learning in Recommender Systems Supplementary material

## Table of Contents

## A    Metrics

In this section, we provide a review of some standard metrics for recommendation system evaluation that we use in our experiments. Recommendation systems return a ranked list of relevant items for each user. We typically care about the top $K$ items returned by the model. There are a number of different methods that evaluate the quality of the predicted list of items versus the ground truth.

Some simple methods include evaluating standard classification metrics such as the precision and recall. There are also some other information retrieval specific metrics that we use: the Normalized Discounted Cumulative Gain and the Mean Average Precision. This section reviews these two metrics.

### A.1    Normalized Discounted Cumulative Gain (NDCG)

To define NDCG we first define its components. The cumulative gain (CG) can be defined as the number of items that a user interacts with that were among the top $K$ predicted by the model:

$$\text{CG@K} = \sum_{k=1}^{K} \mathbb{I}_{[k \in \mathcal{I}^+]}(k) = \sum_{k=1}^{K} G_k. \tag{17}$$

A disadvantage of this metric is that it assigns the same weight to all correct predictions. Predicting 2 items correctly among the top $K$ yields the same CG irrespective of the relative rank of these two items. In general we would especially like the top recommended items to be positive.

To address this we introduce a discount function that weights the cumulative gain from a correctly identified item by its relative ranking. For example, choosing the logarithm function:

$$\text{DCG@K} = \sum_{k=1}^{K} \frac{\mathbb{I}_{[k \in \mathcal{I}^+]}(k)}{\log_2(1+k)} = \sum_{k=1}^{K} \frac{G_k}{\log_2(1+k)}. \tag{18}$$

Finally, to make the DCG score easily comparable across choices of $K$ we may normalize to produce the Normalized DCG (NDCG) by the highest value DCG could attain to produce values in the range $[0, 1]$:

$$\text{NDCG@K} = \frac{\sum_{k=1}^{K} \frac{\mathbb{I}_{[k \in \mathcal{I}^+]}(k)}{\log_2(1+k)}}{\sum_{k=1}^{K} \frac{1}{\log_2(1+k)}} = \frac{\sum_{k=1}^{K} \frac{G_k}{\log_2(1+k)}}{\sum_{k=1}^{K} \frac{1}{\log_2(1+k)}}. \tag{19}$$

We may then report the average NDCG for all users.

### A.2    Mean Average Precision (MAP)

Precision at $K$, denoted as $\text{Prec}(K)$ or prec@$k$, is the fraction of relevant items in the top $K$ items ranked by the model. Similarly to CG from the previous section, a disadvantage of this metric is that it assigns the same weight to all correct predictions. In practice it is more important to identify relevant items at the top of the model's ranking. To address this Average Precision (AP) for user $u$ is used:

$$\text{AP@K}(u) = \frac{1}{|\mathcal{I}_u^+|} \sum_{k=1}^{K} \mathbb{I}_{[k \in \mathcal{I}_U^+]}(k) \, \text{Prec}(k), \tag{20}$$

where $|\mathcal{I}_u^+|$ is the number of relevant items for user $u$. To aggregate this metric we may average over the entire user set $\mathcal{U}$:

$$\text{MAP@K} = \frac{1}{|\mathcal{U}|} \sum_{u \in \mathcal{U}} \text{AP@K}(u). \tag{21}$$

# B  Standard Losses

## B.1  Bayesian Pairwise (BPR) Loss

Reintroducing the notation from the main text, recall that for each user $u$ we have access to the set of items that a user interacts with $\mathcal{I}_u^+ = \{i : (u, i) \in \mathcal{E}\}$ but no explicit set of user dislikes $\mathcal{I}_u^-$. To optimize the model parameters $\Theta$, the Bayesian Pairwise (BPR) loss randomly samples a small number of items with which user $u$ has no observed interactions (Rendle et al., 2009). During training, the overall objective takes the form:

$$\mathcal{L}_{\text{BPR}} := \sum_{(u,i) \in \mathcal{E}, j \in \mathcal{I}_u^-} -\ln \sigma(\hat{y}_{ui} - \hat{y}_{uj}) + \lambda_\Theta \|\Theta\|^2, \tag{22}$$

where $\sigma(x)$ is the standard sigmoid function, and $\lambda_\Theta$ is a regularization parameter. The BPR objective is differentiable, so the model parameters are trainable via standard backpropagation.

Our main contribution, discussed in the methodology section, is a novel approach for selecting $\mathcal{I}_u^-$ that is specifically designed for an incremental learning setting. We also propose a modified triplet loss.

## B.2  Knowledge Distillation (KD) Loss for Incremental Learning in Recommendation

In this section we provide some examples of concrete realizations of standard KD techniques in the context of incremental learning for recommendation systems.

GCNs utilize neural networks to aggregate local information, enabling nodes to benefit from rich contextual data within their neighborhoods and learn generalized representations. Preserving this local neighborhood representation is essential in incrementally trained graph-based recommender systems. To accomplish this, Xu et al. (2020) propose distilling the dot product between the central node embedding and its neighborhood representation. Practically, this can be achieved by applying a KD objective on the dot product between the embedding from time $t-1$ of user $u$, denoted $emb_u^{t-1}$, and the average embedding of the items in its neighborhood $c_{u,\mathcal{N}_u^{t-1}}^{t-1}$ and the dot product of $emb_u^t$ with $c_{u,\mathcal{N}_u^t}^t$. A similar procedure is carried out for item representations and the average embedding of their neighborhoods. Concretely this objective takes the form:

$$\mathcal{L}_{\text{LOCAL}} = \underbrace{\left(\frac{1}{|\mathcal{U}|} \sum_{u \in \mathcal{U}} emb_u^{t-1} \cdot c_{u,\mathcal{N}_u^{t-1}}^{t-1} - emb_u^t \cdot c_{u,\mathcal{N}_u^t}^t\right)^2}_{\mathcal{L}_{\text{KD}}^U} + \underbrace{\left(\frac{1}{|\mathcal{I}|} \sum_{i \in \mathcal{I}} emb_i^{t-1} \cdot c_{i,\mathcal{N}_i^{t-1}}^{t-1} - emb_i^t \cdot c_{i,\mathcal{N}_i^t}^t\right)^2}_{\mathcal{L}_{\text{KD}}^I}, \tag{23}$$

$$\text{where} \quad c_{u,\mathcal{N}_u^t}^t = \frac{1}{|\mathcal{N}_u^{t-1}|} \sum_{i' \in \mathcal{N}_u^{t-1}} emb_{i'}^t \quad \text{and} \quad c_{i,\mathcal{N}_i^t}^t = \frac{1}{|\mathcal{N}_i^{t-1}|} \sum_{u' \in \mathcal{N}_u^{t-1}} emb_{u'}^t.$$

Wang et al. (2021b) propose a contrastive distillation objective. The positive pairs and negative pairs are constructed from the user-item bipartite graph in order to capture the important structural information. The distillation objective is:

$$\mathcal{L}_{sgct} = \mathcal{L}_{KD}^I + \frac{1}{|\mathcal{U}|} \sum_{u \in \mathcal{U}} \frac{-1}{|\mathcal{N}_{UI}^{t-1}(u)|} \sum_{i \in \mathcal{N}_{UI}^{t-1}(u)} \log \frac{\exp\left(\boldsymbol{h}_{u,0}^t \cdot \boldsymbol{h}_{i,0}^{t-1}/\tau\right)}{\sum_{\hat{i} \in \mathcal{D}_{UI}^{t-1}(u)} \exp\left(\boldsymbol{h}_{u,0}^t \cdot \boldsymbol{h}_{i,0}^{t-1}/\tau\right)}, \tag{24}$$

where $\boldsymbol{h}_{u,0}^t$ is the embedding for node $u$ at time $t$, $\mathcal{N}_{UI}^{t-1}(u)$ is the neighborhood set of user node $u$ from the user-item interaction graph at time $t-1$, which provides the positive samples, and $\mathcal{D}_{UI}^{t-1}(u)$ is the collection (union) of positive and negatives samples of the user $u$ generated from the user-item bipartite graph from time $t-1$. $\tau$ is a temperature that adjusts the concentration level.

There are other KD losses such as global structure distillation, which maintains each node's global positional information in the user-item graph (Xu et al., 2020), and personalized distillation objectives tailored to each individual user (Wang et al., 2023).

## C    Dataset Statistics

We evaluate our proposed method empirically on six mainstream recommender system datasets: Gowalla, Yelp, Taobao-14, Taobao-15, Netflix and MovieLens10M. These datasets vary significantly in the total number of interactions, sparsity, average item and user node degrees as well as the time span they cover. Detailed dataset statistics are provided in Appendix Table 4. To simulate an incremental learning setting, each dataset is separated into a base block, containing 60% of the data, followed by four incremental blocks each with 10% of the remaining data, with partitioning applied in chronological order.

Table 4: Statistics of the six datasets used in the experiments.

|  | Gowalla | Yelp | Taobao2014 | Taobao2015 | Netflix | MovieLens10M |
|---|---|---|---|---|---|---|
| Num. edges | 281412 | 942395 | 749438 | 1332602 | 12402763 | 10000054 |
| Num. users | 5992 | 40863 | 8844 | 92605 | 63691 | 71567 |
| Avg. user degrees | 46.96 | 23.06 | 84.74 | 14.39 | 194.73 | 143.11 |
| Num. items | 5639 | 25338 | 39103 | 9842 | 10271 | 10681 |
| Avg. item degrees | 49.90 | 37.19 | 19.17 | 135.40 | 1207.56 | 936.59 |
| Avg. % new users | 2.67 | 3.94 | 1.67 | 2.67 | 4.36 | 4.87 |
| Avg. % new items | 0.67 | 1.72 | 2.60 | 0.22 | 0.72 | 7.12 |
| Time span (months) | 19 | 6 | 1 | 5 | 6 | 168 |

## D    Negative Sampler Baseline Details

In the second set of experiments, we investigate if our user interest shift aware negative reservoir design tailored specifically for incremental learning is effective compared to alternative designs. We investigate how several prominent existing negative sampling strategies perform in incremental learning. The experiment demonstrates that the specialized sampler we propose is better than existing generic negative samplers. The methods we compare to are:

1. **WARP** (Weston et al., 2010): This method randomly samples negatives from the pool of unobserved interactions.

2. **Popularity-based Negative Sampling (PNS)** (Rendle & Freudenthaler, 2014): This method ranks the top negatives and samples them with a fixed parametric distribution, *e.g.*, a geometric.

3. **PinSage Sampler** (Ying et al., 2018a): This negative sampler ranks items based on personalized PageRank and then samples high rank negatives. Note, this is not to be confused with the overall PinSage GNN recommender system backbone — we merely use the negative sampler.

4. **Negative Sample Caching (NS Cache)** (Zhang et al., 2019): This is a knowledge graph negative sampler which we adapt to our setting. It samples negative items and compares them to cache content. The negative samples randomly replace cached entries proportionally to their likelihood following an importance sampling approach. This algorithm has fewer parameters than GAN-based negative samplers, such as KGAN (Cai & Wang, 2018) and IGAN (Wang et al., 2018). Besides, it is fully trainable using back-propagation and has equal or better performance than GAN methods.

5. **Entity Similarity-Based Negative Sampling (ESNS)** (Yao et al., 2022) is a knowledge graph method that we adapt to our setting. It improves negative sampling by considering the semantic similarity between entities and using a shift-based logistic loss function, resulting in higher-quality negative samples and better performance compared to existing methods in link prediction tasks.

6. **DeMix** (Chen et al., 2023) is also adapted from the knowledge graph literature. DeMix works by not only proposing highly scored negative samples but also implements a strategy to alleviate accidental sampling of false negative (but unobserved) interactions.

# E  Algorithms

This section summarizes our algorithm and provides a pseudo-code implementation of the updates of the reservoir at each incremental training block in Alg. 1. Additionally, we also show how the reservoir is used during training to sample negatives in Alg. 2.

---

**Algorithm 1** Updating the Graph-SANE Reservoir

---

**Require:** $\mathcal{R}_t \in \mathbb{R}^{|\mathcal{U}| \times |\mathcal{Q}|}$          ▷ top neg. items per user
**Require:** $\mathbf{H}_t, \mathbf{H}_{t-1} \in \mathbb{N}^{|\mathcal{U}| \times K}$          ▷ histogram of user-item category interactions at times $t, t-1$
**Require:** $\mathbf{C}_t \in \mathbb{R}^{|\mathcal{I}| \times K}$          ▷ item to category one-hot map at time $t$
 1: $\mathbf{M}_t \in \mathbb{R}^{|\mathcal{U}| \times |\mathcal{Q}|} \leftarrow 0$          ▷ stores probability of sampling neg. item $i$ for user $u$
 2: **for** $u = 0; u \leq |\mathcal{U}|; u++$ **do**
 3:      **for** $i = 0; i \leq |\mathcal{Q}|; i++$ **do**
 4:          $\hat{\boldsymbol{\theta}}_{u,t} \in \mathbb{R}^K \leftarrow$ update sampling params from eq. equation 10
 5:          $C \leftarrow \arg\max \mathbf{C}_t[i, :]$          ▷ get category $i$ belongs to
 6:          $\mathbf{M}_t[u, i] \leftarrow p(n_i = i | i \in C, \hat{\boldsymbol{\theta}}_{u,t})$          ▷ from eq. equation 11
 7:      **end for**
 8: **end for**

---

**Algorithm 2** Incremental Training with Graph-SANE Reservoir

---

**Require:** $\Theta_{t-1}$          ▷ params of model from block $t-1$
**Require:** $f$, max_epochs          ▷ refresh rate of reservoir, max epochs
**Require:** $\mathcal{R}_t \in \mathbb{R}^{|\mathcal{U}| \times |\mathcal{Q}|}$          ▷ SANE reservoir
**Require:** $N_1, N_2$          ▷ num. random negatives, reservoir samples
 1: N_ITER $\leftarrow 0$
 2: **repeat**
 3:      **if** N_ITER mod $f == 0$ **then**
 4:          Update item cluster memberships using eq. equation 15
 5:          Update $\mathcal{R}_t$ using Algorithm 1
 6:      **end if**
 7:      Draw a batch of incremental data $\mathcal{B} \subseteq \{(u, i) \in \mathcal{E}_{[t-1,t)}\}$
 8:      Draw $N_1$ random negatives $\mathcal{I}_u^-$
 9:      Sample $N_2$ reservoir negatives $\mathcal{I}_u^{*-}$
10:      Compute loss components from eq. equation 16
11:      Update parameters $\Theta$
12:      N_ITER $\leftarrow$ N_ITER $+ 1$
13: **until** convergence

---

# F  Case Study Details

To conduct this case study we chose a clustering algorithm that differs from the one used in our method. This is done to provide an alternative estimate of high shift users that is not part of our model. Note that the user shift indicator, *i.e.* the technique to identify high interest shift users, we use follows the process introduced by Wang *et al.* (Wang et al., 2023). For completeness we explicitly list the steps taken by Wang *et al.* (Wang et al., 2023):

1. Apply K-means on item embeddings at time block $t$ obtained from the SGCT model to identify $K$ clusters.

2. Obtain $\tilde{\boldsymbol{I}}^t \in \mathbb{R}^{U \times K}$ by counting number of items from each category a user interacts with for all $t$.

3. Normalize $\tilde{\boldsymbol{I}}^t$ to calculate $\boldsymbol{I}_k = \tilde{\boldsymbol{I}}^t_k / \sum_{k' \in K} \tilde{\boldsymbol{I}}^{t'}_{k}$.

4. Obtain interest shift indicator: $ISS_u = \frac{1}{K} \sum_{k=1}^{K} ||\boldsymbol{I}_{u,k}^t - \boldsymbol{I}_{u,k}^{t-1}||^2$.

5. We define users with top 15% interest shift indicator as the *high shift user* set.

6. Calculate average recall for the high shift users using all the different negative samplers on top of SGCT.

## G  Loss Ablation Study & Sensitivity Analysis

### G.1  Ablation study

In this section we conduct a loss ablation study on our proposed objective, as well as sensitivity studies on key components of our method: the choice of specific clustering algorithm, the number of clusters of our clustering algorithm as well as the size of our proposed negative reservoir. The dataset choice for the ablation and sensitivity experiments can be explained by the fact that Netflix, which is our biggest dataset, is prohibitively big to run many experiments on as it takes over 2 days per experiment so we opt for Yelp and Taobao14 as they have the most representative number of user and items compared to the average dataset (they are neither the biggest nor the smallest).

Table 5: Loss components ablation on Taobao14 and Yelp for **SGCT-SANE** (Recall@20). We check the impact of removing each loss term from our overall proposed optimization objective. As we can see, removing any of our proposed components impacts performance and/or end-to-end trainability.

| Method | $\mathcal{L}_{\text{BPR}}$ | $\mathcal{L}_{\text{KD}}$ | $\mathcal{L}_{\text{SANE}}$ | $\mathcal{L}_{\text{KL}}$ | End-to-End Trainable | Taobao14 | Yelp |
|---|---|---|---|---|---|---|---|
| Fine Tune | ✓ | ✗ | ✗ | ✗ | Yes | 0.0153 | 0.0661 |
| SGCT | ✓ | ✓ | ✗ | ✗ | Yes | 0.0160 | 0.0668 |
| SGCT-hard-cluster | ✓ | ✓ | ✓ | ✗ | No | **0.0178** | 0.0845 |
| SGCT-SANE (ours) | ✓ | ✓ | ✓ | ✓ | Yes | **0.0178** | **0.0857** |

Table 6: Loss components ablation on Taobao14 and Yelp for **LWCKD-SANE** (Recall@20). As we can see, removing any of our proposed components impacts performance and/or end-to-end trainability.

| Method | $\mathcal{L}_{\text{BPR}}$ | $\mathcal{L}_{\text{KD}}$ | $\mathcal{L}_{\text{SANE}}$ | $\mathcal{L}_{\text{KL}}$ | End-to-End Trainable | Taobao14 | Yelp |
|---|---|---|---|---|---|---|---|
| Fine Tune | ✓ | ✗ | ✗ | ✗ | Yes | 0.0153 | 0.0661 |
| LWCKD | ✓ | ✓ | ✗ | ✗ | Yes | 0.0176 | 0.0679 |
| LWCKD-hard-cluster | ✓ | ✓ | ✓ | ✗ | No | 0.0185 | 0.0857 |
| LWCKD-SANE (ours) | ✓ | ✓ | ✓ | ✓ | Yes | **0.0188** | **0.0898** |

Our loss ablation study validates each term in the proposed loss. Concretely, we check the impact of removing each loss term from our overall proposed optimization objective. As we can see in Tables 5 and 6 removing any of our proposed components impacts performance and/or end-to-end trainability.

### G.2  Clustering Algorithm Sensitivity

In this sensitivity study we check if an ad-hoc training scheme where we cluster the items using K-means between epochs can produce similar results to our approach, which uses an end-to-end structural based clustering method (Bo et al., 2020). In Table 8 observe that our method performs within 1-3% with either clustering algorithm, thus validating the low sensitivity towards clustering algorithm selection. We obtain similar results when we vary the dataset choice and the number of clusters. The attributed clustering algorithm maintains end-to-end trainability so it converges faster than K-means.

In the second sensitivity study, shown in Table 9, we conduct an analysis on the number of clusters used in our method. As we can see, once the number of clusters reaches a sufficient number ($\sim 10$), the performance remains stable.

Thirdly, in Table 10, we conduct a sensitivity analysis on the size of the user negative reservoir $|\mathcal{Q}|$. As we can see, the method is not sensitive to the size of the reservoir. This implies that introducing even a few hard negatives in the incremental training can be sufficient to improve performance.

Table 7: Sensitivity to clustering algorithm choice in Yelp. Results demonstrate low sensitivity to specific algorithm choice.

| Methods | Inc. 1 | Inc. 2 | Inc.3 | Avg. |
|---|---|---|---|---|
| GraphSAIL+SANE (K-means) | 0.0877 | 0.0871 | 0.0791 | 0.0846 |
| GraphSAIL+SANE (end-to-end - ours) | 0.0939 | 0.0842 | 0.0791 | 0.0857 |
| SGCT+SANE (K-means) | 0.0946 | 0.0858 | 0.0732 | 0.0845 |
| SGCT+SANE (end-to-end - ours) | 0.0966 | 0.0877 | 0.0744 | 0.0857 |
| LWC-KD+SANE (K-means) | 0.0931 | 0.0853 | 0.0787 | 0.0857 |
| LWC-KD+SANE (end-to-end - ours) | 0.0924 | 0.0855 | 0.0798 | 0.0859 |

Table 8: Sensitivity to clustering algo. choice in Taobao14. Results demonstrate low sensitivity to specific algorithm choice.

| Methods | Inc. 1 | Inc. 2 | Inc.3 | Avg. |
|---|---|---|---|---|
| GraphSAIL+SANE (K-means) | 0.0222 | 0.0139 | 0.0165 | 0.0175 |
| GraphSAIL+SANE (end-to-end) | 0.0231 | 0.0131 | 0.0150 | 0.0171 |
| SGCT+SANE (K-means) | 0.0228 | 0.0154 | 0.0153 | 0.0178 |
| SGCT+SANE (end-to-end) | 0.0224 | 0.0136 | 0.0173 | 0.0178 |
| LWC-KD+SANE (K-means) | 0.0228 | 0.0174 | 0.0154 | 0.0185 |
| LWC-KD+SANE (end-to-end) | 0.0222 | 0.0188 | 0.0156 | 0.0188 |

Table 9: Sensitivity analysis for the cluster number K. We present Recall@20 results on the Gowalla and Taobao2015 datasets for SGCT-SANE and LWC-KD-SANE. Results demonstrate low sensitivity to number of clusters.

| Distillation Strategies | Dataset | K | Inc 1 | Inc 2 | Inc 3 | Avg. Recall@20 |
|---|---|---|---|---|---|---|
| | | 5 | 0.0240 | 0.0133 | 0.0127 | 0.0167 |
| | | 10 | 0.0224 | 0.0136 | 0.0173 | 0.0178 |
| | SGCT-SANE | 15 | 0.0237 | 0.0143 | 0.0143 | 0.0174 |
| | | 20 | 0.0237 | 0.0154 | 0.0165 | 0.0185 |
| Taobao14 | | 25 | 0.0234 | 0.0156 | 0.0166 | 0.0185 |
| | | 5 | 0.0265 | 0.0121 | 0.0150 | 0.0179 |
| | | 10 | 0.0222 | 0.0188 | 0.0155 | 0.0188 |
| | LWC-KD-SANE | 15 | 0.0251 | 0.0138 | 0.0161 | 0.0183 |
| | | 20 | 0.0247 | 0.0128 | 0.0145 | 0.0173 |
| | | 25 | 0.0247 | 0.0141 | 0.0162 | 0.0183 |

Table 10: Sensitivity analysis on the size of the user negative reservoir $|\mathcal{Q}|$. We present Recall@20 results on the Gowalla and Taobao2015 datasets for SGCT-SANE and LWC-KD-SANE. Results demonstrate low sensitivity to negative reservoir size.

| Distillation Strategies | Dataset | $|\mathcal{Q}|$ | Inc 1 | Inc 2 | Inc 3 | Avg. Recall@20 |
|---|---|---|---|---|---|---|
| | | 50 | 0.0237 | 0.0156 | 0.0160 | 0.0184 |
| | SGCT-SANE | 100 | 0.0224 | 0.0136 | 0.0173 | 0.0178 |
| | | 300 | 0.0254 | 0.0140 | 0.0160 | 0.0185 |
| Taobao14 | | 50 | 0.0238 | 0.0140 | 0.0162 | 0.0180 |
| | LWC-KD-SANE | 100 | 0.0222 | 0.0188 | 0.0155 | 0.0188 |
| | | 300 | 0.0246 | 0.0133 | 0.0180 | 0.0186 |

# H    Hyperparameter Settings

Our method is implemented in TensorFlow. The backbone graph neural network is the MGCCF (Sun et al., 2019) trained using the hyperparameters shown in Table 11. Incremental learning methods are not used during the base block training so the loss during the base block is only $\mathcal{L}_{\text{BPR}}$, i.e., (no $\mathcal{L}_{\text{KD}}$, $\mathcal{L}_{\text{SANE}}$, $\mathcal{L}_{\text{KL}}$ components).

Table 11: Hyperparameters of our model on all benchmarks.

| Hyperparameter | Value |
|---|---|
| Min Epochs Base Block | 10 |
| Min Epochs Incremental | 3 |
| Max Epochs Base Block | N/A |
| Max Epochs Incremental | 15 |
| Early Stopping Patience | 2 |
| Batch size | 64 |
| Optimizer | Adam |
| Cache Update Frequency $f$ | 2 epochs |
| Cache Size per user $|\mathcal{Q}|$ | 100 |
| Learning rate (max) | 5e-4 |
| Dropout | 0.2 |
| Losses | KD, BPR, SANE, KL |
| GNN Num Layers ($L$) | 2 |
| Num Clusters ($K$) | 10 |
| Embedding dimensionality | 128 |
| Augmentations | NONE |

# I Experimental Results per Incremental Training Block

In this section we provide a more detailed version of the results from Tab. 1 with a breakdown of the performance for each incremental block.

Table 12: Comparison (Recall@20) of baselines and 3 recent knowledge distillation algorithms with our SANE reservoir, our methods are accompanied by a star (*). Best performers are in bold, second best are underlined. Blue cell colors indicate improvement of a particular method when our reservoir is introduced.

| Yelp | | | | | | Netflix | | | | |
|---|---|---|---|---|---|---|---|---|---|---|
| Model | Inc 1 | Inc 2 | Inc 3 | Average | Δ% | Inc 1 | Inc 2 | Inc 3 | Average | Δ% |
| Fine Tune | 0.0705 | 0.0638 | 0.0640 | 0.0661 | 0.0 | 0.1092 | 0.1041 | 0.0977 | 0.1036 | 0.0 |
| LSP_s | 0.0722 | 0.0661 | 0.0644 | 0.0676 | 2.3 | 0.1173 | 0.1136 | 0.1076 | 0.1128 | 8.7 |
| Uniform | 0.0718 | 0.0635 | 0.0610 | 0.0654 | -1.1 | 0.1018 | 0.1055 | 0.0800 | 0.0957 | -7.6 |
| Inv_degree | 0.0727 | 0.0699 | 0.0605 | 0.0677 | 2.4 | 0.1000 | 0.1050 | 0.0820 | 0.0957 | -7.6 |
| GraphSAIL | 0.0674 | 0.0617 | 0.0625 | 0.0639 | -3.3 | 0.1163 | 0.1023 | 0.0968 | 0.1051 | 1.4 |
| GraphSAIL-SANE* | 0.0939 | 0.0842 | 0.0791 | 0.0857 | 29.7 | 0.1153 | 0.1091 | 0.01014 | 0.1086 | 4.8 |
| SGCT | 0.0740 | 0.0656 | 0.0608 | 0.0668 | 1.06 | 0.1166 | 0.1161 | 0.1077 | 0.1135 | 9.6 |
| SGCT-SANE* | 0.0966 | 0.0877 | 0.0744 | 0.0862 | 30.4 | 0.1182 | 0.1164 | 0.1120 | 0.1155 | 11.5 |
| LWC-KD | 0.0739 | 0.0661 | 0.0637 | 0.0679 | 2.7 | 0.1185 | 0.1170 | 0.1071 | 0.1142 | 10.2 |
| LWC-KD-SANE* | 0.0970 | 0.0891 | 0.0834 | 0.0898 | **35.9** | 0.1192 | 0.1196 | 0.1157 | 0.1182 | **14.1** |
| Taobao14 | | | | | | Taobao15 | | | | |
| Model | Inc 1 | Inc 2 | Inc 3 | Average | Δ% | Inc 1 | Inc 2 | Inc 3 | Average | Δ% |
| Fine Tune | 0.0208 | 0.0112 | 0.0138 | 0.0153 | 0.0 | 0.0933 | 0.0952 | 0.0965 | 0.0950 | 0.0 |
| LSP_s | 0.0213 | 0.0106 | 0.0138 | 0.0152 | -0.7 | 0.0993 | 0.0952 | 0.0957 | 0.0968 | 1.9 |
| Uniform | 0.0195 | 0.0127 | 0.0148 | 0.0157 | 2.6 | 0.0988 | 0.0954 | 0.1004 | 0.0982 | 3.4 |
| Inv_degree | 0.0228 | 0.0140 | 0.0159 | 0.0175 | 14.6 | 0.0991 | 0.0977 | 0.1000 | 0.0989 | 4.2 |
| GraphSAIL | 0.0222 | 0.0105 | 0.0139 | 0.0155 | 1.31 | 0.0959 | 0.0959 | 0.0972 | 0.0963 | 1.4 |
| GraphSAIL-SANE* | 0.0231 | 0.0131 | 0.0150 | 0.0171 | 11.8 | 0.1114 | 0.1087 | 0.1121 | 0.1107 | 16.5 |
| SGCT | 0.0240 | 0.0092 | 0.0148 | 0.0160 | 1.74 | 0.1030 | 0.0983 | 0.0984 | 0.0999 | 5.2 |
| SGCT-SANE* | 0.0224 | 0.0136 | 0.0173 | 0.0178 | 16.3 | 0.1117 | 0.1129 | 0.1138 | 0.1128 | **18.7** |
| LWC-KD | 0.0254 | 0.0119 | 0.0156 | 0.0176 | 15.3 | 0.1039 | 0.1022 | 0.1029 | 0.1030 | 8.4 |
| LWC-KD-SANE* | 0.0222 | 0.0188 | 0.0155 | 0.0188 | **22.9** | 0.1106 | 0.1108 | 0.1128 | 0.1114 | 17.2 |
| Gowalla-20 | | | | | | MovieLens10M | | | | |
| Model | Inc 1 | Inc 2 | Inc 3 | Average | Δ% | Inc 1 | Inc 2 | Inc 3 | Average | Δ% |
| Fine Tune | 0.1412 | 0.1637 | 0.2065 | 0.1705 | 0.0 | 0.0923 | 0.0904 | 0.0957 | 0.0918 | 0.0 |
| LSP_s | 0.1512 | 0.1741 | 0.2097 | 0.1783 | 4.57 | 0.0912 | 0.0923 | 0.0941 | 0.0925 | 0.76 |
| Uniform | 0.1480 | 0.1653 | 0.2051 | 0.1728 | 1.34 | 0.0874 | 0.0879 | 0.0902 | 0.0885 | -3.59 |
| Inv_degree | 0.1483 | 0.1680 | 0.2001 | 0.1738 | 1.93 | 0.0906 | 0.0911 | 0.0928 | 0.0915 | -0.32 |
| GraphSAIL | 0.1529 | 0.1823 | 0.2195 | 0.1849 | 8.44 | 0.0867 | 0.0887 | 0.0917 | 0.0890 | -3.05 |
| GraphSAIL-SANE* | 0.1646 | 0.1907 | 0.2221 | 0.1925 | 12.9 | 0.0887 | 0.0915 | 0.0961 | 0.0921 | 0.33 |
| SGCT | 0.1588 | 0.1815 | 0.2207 | 0.1870 | 9.68 | 0.0894 | 0.0902 | 0.0960 | 0.0919 | 0.11 |
| SGCT-SANE* | 0.1611 | 0.1843 | 0.2237 | 0.1897 | 11.3 | 0.0927 | 0.0932 | 0.0980 | 0.0946 | 3.05 |
| LWC-KD | 0.1639 | 0.1921 | 0.2368 | 0.1977 | **15.9** | 0.0902 | 0.0910 | 0.0973 | 0.0928 | 1.09 |
| LWC-KD-SANE* | 0.1698 | 0.1835 | 0.2173 | 0.1881 | 10.3 | 0.0928 | 0.0953 | 0.1012 | 0.0964 | **5.01** |

## J  Dataset Preprocessing - Data Block Definition

For all data sets we follow the setup from literature (Xu et al., 2020; Ahrabian et al., 2021). We divide the data in chronological order into a base block and incremental blocks. The base block contains 60% of the data. The remaining 40% of the data is evenly separated into four incremental blocks. Note that during training the blocks can be further subdivided to allow for a validation set. In our setting for each block we reserve 5% of the data as a validation dataset following from the baselines (Xu et al., 2020; Ahrabian et al., 2021). Note that this does not induce data leakage as all splits are in chronological order. For a visualization see Fig. 5.

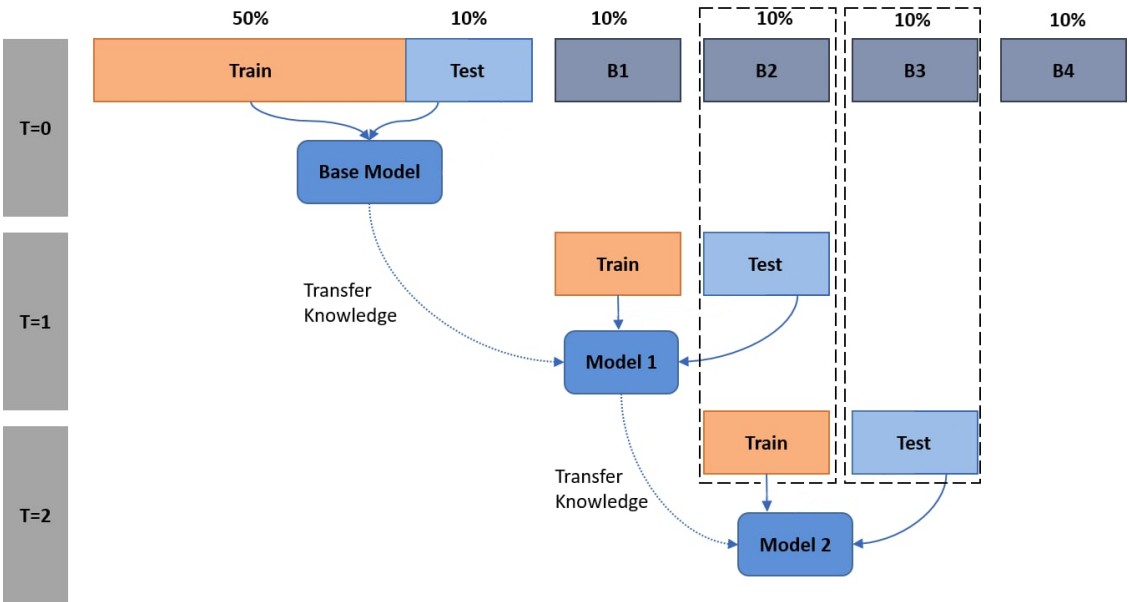

Figure 5: Data is separated to a base block with 60% of the data and 4 incremental blocks, each with 10% of the remaining data.

# K   Practical Recommendation for Model Deployment

Tuning the framework hyperparameters can be achieved in similar fashion to the setup in Fig. 5. Here, we again split the data into temporally sorted blocks. The base block contains 60% of the data. The remaining 40% of the data is evenly separated into four incremental blocks. During training, when block $t$ is used as the training set, the first half of block $t + 1$ is used as the validation set and the second half is used as the testing set (Figure 6). Note this setup prevents data leakage as the test data are all from interactions that occur after the validation set. For example, if a block contains a 10 days worth of data, the validation dataset would contain days 1-5 and the test set days 6-10.

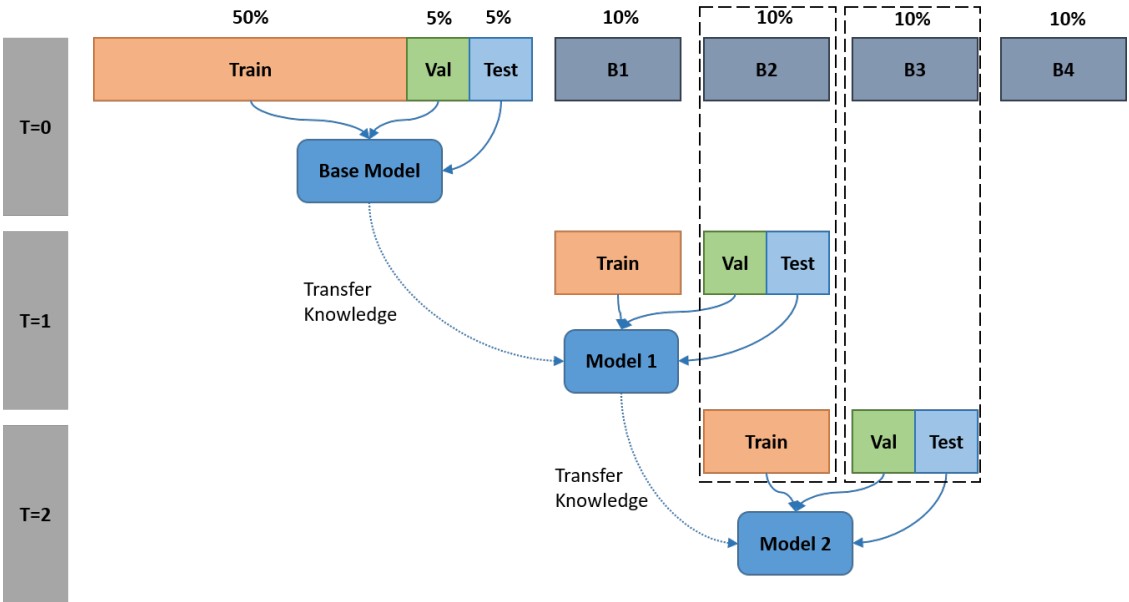

Figure 6: Incremental learning hyperparameter tuning setup. Note this setup prevents data leakage as the test data are all from interactions that occur after the validation data. For example, if a block contains a 10 days worth of data, the validation dataset would contain days 1-5 and the test set days 6-10.

To deploy GraphSANE on top of an existing recommendation system, we recommend the following steps:

1. Accumulate data equal to 10 times the incremental block size. For example, if the model is expected to be updated daily we recommend accumulating 10 days worth of data for hyperparameter tuning.

2. Split the data chronologically in blocks as shown in Fig. 6. The reason that we put validation sets in the data immediately following the block is the we wish to create a back-testing style setup where the hyperpameters are tuned based on the typical distribution shift between the train and test steps. Note that since the blocks are chronologically ordered this does not leak any interaction from the test set to the validation set. This placement of the validation set is a practice adopted form the literature (Xu et al., 2020; Ahrabian et al., 2021).

3. Train a base GNN model on the base block data. Note that any base backbone recommendation system model, such as MGCCF (Sun et al., 2019) or LightGCN (He et al., 2020b) has its own hyperparameters which can be tuned on the validation portion of the base block. These base GNN architectures have their own hyperparameters that are not related to our incremental learning framework so we omit further discussion on how to tune the base models and focus on how to shift hyperparameters related to our method.

4. Tune $\beta$ parameter by checking the training loss of the first incremental block. It suffices to select $\beta$ values such that at the start of training the KL loss component of the loss is in the same scale as

the triplet loss component. Note that the source code we provide tracks these components so this process is easy to perform.

5. For the number of clusters $K$ and the size of the reservoir $|Q|$ we recommend running a grid search. Note that since these are two hyperparameters we may run a search over the values we used in the sensitivity tests we run in Tables 9 and 10. Note even though we run a grid search in our experiments, the parameter tuning process could be further optimized using Bayesian optimization hyperparameter tuning tools such as Optuna [2].

---

[2]https://optuna.org/

