# OpenReview forum: "Personalized Negative Reservoir for Incremental Learning in Recommender Systems"
_TMLR — Accepted by TMLR_

### Review · Reviewer_UQg6 · 2024-11-10

**Summary Of Contributions:**

This paper proposes a personalized negative reservoir strategy for incremental learning in recommender systems, addressing the issue of catastrophic forgetting while adapting to changes in user preferences. The authors introduce a novel negative sampling strategy tailored for incremental learning, validated through integration with three state-of-the-art (SOTA) models. The key contributions include a mathematical derivation of a personalized negative reservoir, integration with incremental recommendation frameworks, and experimental results verified on five large-scale datasets.

**Audience:**

Yes

**Broader Impact Concerns:**

No concerns in this regard.

**Claims And Evidence:**

Yes

**Requested Changes:**

1. Provide detailed information on how the test set is determined in the incremental learning setup of recommender system datasets.

2. Further discuss how the proposed negative reservoir strategy enhances the stability of the incremental learning system, helping to address the issue of catastrophic forgetting.

3. Replace $\mathbf{H}_{u,t-1}$ in Eq.7 with prefix sum or sliding average to perform an ablation study.

4. Include the Joint Training method.

5. Include a baseline that learns only on incremental blocks.

6. Add a comparison with *at least one* recent negative sampling methods.

**Strengths And Weaknesses:**

**Strengths**:

1. The paper addresses the important but underexplored issue of negative sampling in incremental learning, which is crucial for the performance of recommender systems.

2. The proposed method shows significant improvements across multiple metrics (e.g., Recall@20), especially when combined with SOTA models.

3. The personalized negative reservoir strategy effectively handles user interest shifts, making it highly valuable for practical recommender system applications.

**Weaknesses**:

1. The benchmark used in this paper does not clearly specify how the test set is defined, which is important for interpreting the assumptions in Section 5.2 and understanding the reasons for performance improvements. My understanding is that the incremental split dataset is sorted by time, and after each incremental block is learned, all the remaining data including the most recent training data should be used as the test set.

2. SANE improves plasticity in incremental learning by considering user interest shifts, but a thorough incremental learning framework also needs to consider stability, as catastrophic forgetting has a detrimental impact on system performance. I suggest further explaining how the proposed method contributes to system stability.

3. The generalizability of calculating interest shifts based solely on user interactions from the previous time block is questionable. While this approach is reasonable in scenarios with rapid user interest changes, it may not be applicable to all incremental cases. It is recommended to add an ablation study by replacing $\mathbf{H}_{u,t-1}$ in Eq.7 with prefix sum or sliding average and to provide related discussions to support the stability and generalizability of the SANE system.

4. The paper lacks the comparison with the Oracle method Joint Training in the incremental learning. Further, if user interest shifts are rapid and significant, would learning only on the incremental blocks achieve good results? I suggest adding a relevant comparison.

5. The choice of baseline negative sampling methods could be more comprehensive. It is recommended to refer to the latest survey[1] and add a comparison with *at least one* recent negative sampling technique, such as ESNS[2] or DeMix[3], to strengthen the evaluation.

**Minor Question**

* In Section 6.3, the statement "Furthermore, as summarized in Fig.3, LWC-KD-SANE is the top-performing method overall" seems inconsistent with the average rank shown in Fig.3, where SGCT-SANE appears to perform better.

**Reference**:

[1] Madushanka, T., & Ichise, R. (2024). Negative Sampling in Knowledge Graph Representation Learning: A Review. ArXiv, abs/2402.19195.

[2] Yao, N., Liu, Q., Li, X., Yang, Y., & Bai, Q. (2022). Entity Similarity-Based Negative Sampling for Knowledge Graph Embedding. Pacific Rim International Conference on Artificial Intelligence.

[3] Chen, X., Zhang, W., Yao, Z., Chen, M., & Tang, S. (2023). Negative Sampling with Adaptive Denoising Mixup for Knowledge Graph Embedding. International Workshop on the Semantic Web.

---

> ### Author Response · Authors · 2024-12-28
>
> We would like to thank the reviewer for the thorough review and the insightful feedback. Our detailed response is below. If you have any additional concerns, please reach out to us to discuss them.

---

> ### Author Response · Authors · 2024-12-28
> **Experimental Setup**
>
> > **Experimental Setup** The benchmark used in this paper does not clearly specify how the test set is defined, which is important for interpreting the assumptions in Section 5.2 and understanding the reasons for performance improvements. My understanding is that the incremental split dataset is sorted by time, and after each incremental block is learned, all the remaining data including the most recent training data should be used as the test set.
>
> The description of the data split process is essentially as the reviewer described. In the revision we clarify how the incremental data blocks are defined in newly added Appendix J and summarized via a schematic in Figure 4. Note that all our datasets are processed and set up in the exact same way as the baselines and prior works to ensure fair comparison.

---

> ### Author Response · Authors · 2024-12-28
> **Stability**
>
> > **Stability** SANE improves plasticity in incremental learning by considering user interest shifts, but a thorough incremental learning framework also needs to consider stability, as catastrophic forgetting has a detrimental impact on system performance. I suggest further explaining how the proposed method contributes to system stability.
>
> We agree with the reviewer that achieving a good balance between plasticity and stability is a key aspect of developing a good continual learning framework. In this paper we argue that a core limitation of prior works in continual learning for graph recommender systems is the excessive emphasis on stability (positive interaction reservoir, knowledge distillation based techniques). Thus these approaches have failed to provide sufficient balance with respect to plasticity. Therefore, our approach is proposed as an add-on on top of the stability based incremental learning techniques to correct the imbalance and allow the incremental learning framework to have high plasticity for users that exhibit distribution shift in preferences while preventing forgetting for users with stable preferences.

---

> ### Author Response · Authors · 2024-12-28
> **Prefix Sum Ablation**
>
> > **Prefix Sum Ablation** The generalizability of calculating interest shifts based solely on user interactions from the previous time block is questionable. While this approach is reasonable in scenarios with rapid user interest changes, it may not be applicable to all incremental cases. It is recommended to add an ablation study by replacing $\mathbf{H}_{u, t-1}$ in Eq.7 with prefix sum or sliding average and to provide related discussions to support the stability and generalizability of the SANE system.
>
> We agree with the reviewer that the approach as proposed is more effective for users with rapid interest shift. We emphasize that this is our exact intention, as our approach serves as an add-on that specializes in handling high-shift users on top of existing incremental learning methods that are applicable to stable users. This is also verified in the case study in Section 6.6 where we see that the negative reservoir performs disproportionately well on the high interest shift user subset.
>
> Our approach takes the first step and proposes a basic framework that works well for rapidly shifting user interests. We believe that exploration of more elaborate user shift mechanisms to detect slow shifts over longer horizons would be an exciting future direction for research to improve the stability of negative reservoirs.

---

> ### Author Response · Authors · 2024-12-28
> **Additional Baselines**
>
> > **Baseline 1** The paper lacks the comparison with the Oracle method Joint Training in the incremental learning. Further, if user interest shifts are rapid and significant, would learning only on the incremental blocks achieve good results? I suggest adding a relevant comparison.
>
> We thank the reviewer for bringing this method up. Unfortunately, after searching online we have not been able to identify the paper that details this method. Could the reviewer kindly point us in the correct direction by providing a full bibliographical reference?
>
> > **Baseline 2** The choice of baseline negative sampling methods could be more comprehensive. It is recommended to refer to the latest survey [1] and add a comparison with at least one recent negative sampling technique, such as ESNS[2] or DeMix[3], to strengthen the evaluation.
> [1] Madushanka, T., & Ichise, R. (2024). Negative Sampling in Knowledge Graph Representation Learning: A Review. ArXiv, abs/2402.19195.
> [2] Yao, N., Liu, Q., Li, X., Yang, Y., & Bai, Q. (2022). Entity Similarity-Based Negative Sampling for Knowledge Graph Embedding. Pacific Rim International Conference on Artificial Intelligence.
> [3] Chen, X., Zhang, W., Yao, Z., Chen, M., & Tang, S. (2023). Negative Sampling with Adaptive Denoising Mixup for Knowledge Graph Embedding. International Workshop on the Semantic Web.
>
> We thank the reviewer for pointing out newer works in the negative sampling literature. We have added comparison the two methods the reviewer recommended (DeMix and ESNS) with our approach still remaining the best performer. In the revision we update Table 3 by adding two new rows for each dataset.
>
> Results for Yelp:
>
> | Methods            |  Inc 1 |  Inc 2 |  Inc 3 |  Avg.  |
> |--------------------|:------:|:------:|:------:|:------:|
> | SGCT+[Warp]        | 0.0740 | 0.0656 | 0.0608 | 0.0668 |
> | SGCT+[PinSage]     | 0.0794 | 0.0663 | 0.0651 | 0.0703 |
> | SGCT+[PNS]         | 0.0933 | 0.0798 | 0.0748 | 0.0827 |
> | SGCT+[NS Cache]    | 0.0794 | 0.0681 | 0.0670 | 0.0715 |
> | SGCT+[DeMix]       | 0.0783 | 0.0740 | 0.0694 | 0.0739 |
> | SGCT+[ESNS]        | 0.0786 | 0.0754 | 0.0690 | 0.0743 |
> | SGCT+[SANE] (ours) | 0.0966 | 0.0877 | 0.0744 | **0.0862** |
>
> Results for Taobao14:
>
> | Methods            |  Inc 1 |  Inc 2 |  Inc 3 |  Avg.  |
> |--------------------|:------:|:------:|:------:|:------:|
> | SGCT+[Warp]        | 0.0240 | 0.0092 | 0.0148 | 0.0160 |
> | SGCT+[PinSage]     | 0.0241 | 0.0099 | 0.0151 | 0.0164 |
> | SGCT+[PNS]         | 0.0220 | 0.0114 | 0.0113 | 0.0149 |
> | SGCT+[NS Cache]    | 0.0237 | 0.0124 | 0.0121 | 0.0165 |
> | SGCT+[DeMix]       | 0.0196 | 0.0097 | 0.0113 | 0.0135 |
> | SGCT+[ESNS]        | 0.0218 | 0.0113 | 0.0128 | 0.0153 |
> | SGCT+[SANE] (ours) | 0.0224 | 0.0136 | 0.0173 | **0.0178** |
>
> > **Baseline 3** Include a baseline that learns only on incremental blocks.
>
> Thank you for the recommendation. As discussed in the response to reviewer 2utX Fine Tune is an online method that is initialized from weights of previous block and learns only from data of the incremental block. We interpret that as "learning only on incremental block data" and show that incremental learning approaches that explicitly attempt to prevent catastrophic forgetting outperform Fine Tune.
>
> Another interpretation of "learning only on incremental block data" could be a baseline where the weights are initialized randomly with the training starting completely from scratch for each block. While this would provide an additional comparison we would argue that this is not likely to be a good baseline as it would completely discard all knowledge gained from previous data blocks which is the majority of the data.

---

> ### Author Response · Authors · 2024-12-28
> **Minor Question**
>
> > **Minor Question** In Section 6.3, the statement "Furthermore, as summarized in Fig.3, LWC-KD-SANE is the top-performing method overall" seems inconsistent with the average rank shown in Fig.3, where SGCT-SANE appears to perform better.
>
> We thank the reviewer for the careful reading of our paper and spotting this plotting error. In a revision we have corrected this issue and updated the figure to correct SGCT-SANE and LWC-KD-SANE relative ranks.

---

> ### Comment · Reviewer_UQg6 · 2024-12-29
>
> Thank you for addressing my concerns. A clear illustration of the dataset setup is crucial for understanding the proposed method, and the added baseline experiments further validate the superiority of the proposed approach.
>
> As for my **W4** and **R4**, Joint Training refers to a scenario allowing access to blocks $0, \dots, t-1$ while training block $t$, and training on all blocks' data jointly. It disregards the setup of incremental learning constraints. In conventional continual learning settings, Joint Training is considered the performance upper bound. However, in the context of this study, Joint Training may be suboptimal due to the rapid shift in user interests. Validating this assumption would enhance the persuasiveness of the proposed method and provide further inspiration for the continual learning community.

---

> > ### Author Response · Authors · 2024-12-29
> >
> > We are glad that the revised manuscript addresses your concerns.
> >
> > Thank you for clarifying the question regarding Joint Training. We actually have a method that follows the scenario you outlined.
> >
> > In Table 1 we compare to Uniform random sampling of all past and present interactions. That is, each batch contains data both from the new block and all previously seen blocks. Additionally, the Inverse Degree method from Ahrabian et al. 2021 builds on this idea and samples data from old blocks with more elaborate distribution (rather than uniform).
> >
> > As the reviewer suggested, looking at the results, while this approach can prevent catastrophic forgetting and improves over a few naive baselines in our setting Joint Training is suboptimal due to the rapid shift in user interests and does not achieve top results compared to more elaborate methods.
> >
> > We will add a paragraph to explain this in the camera ready version of the manuscript.

---

> ### Comment · Reviewer_UQg6 · 2024-12-29
>
> Thank you for your response. Including a discussion on the rapidly changing user interests in incremental learning for recommendation systems, rather than focusing on preventing catastrophic forgetting as in traditional continual learning, would significantly enhance the contribution of this work.

---

> > ### Author Response · Authors · 2024-12-29
> >
> > Thank you for the feedback. As the reviewer requested, we will add discussion indicating that the focus of our work is on accounting for user interest shifts rather addressing the problem from a traditional continual learning perspective that focuses on preventing catastrophic forgetting in the final version of the manuscript.

---

### Review · Reviewer_r8Jp · 2024-12-02

**Summary Of Contributions:**

The paper, titled Personalized Negative Reservoir for Incremental Learning in Recommender Systems, proposes a novel strategy to address the challenge of maintaining efficient and accurate recommendation systems as user interactions evolve over time. The proposed framework can be intergrated with existing graph-based recommendation systems and incremental learning models and it is used to improve performance of state-of-the-art Graph Incremental Learning methods (GraphSAIL, SGCT, and LWC-KD) across five datasets.

**Audience:**

Yes

**Broader Impact Concerns:**

No issues.

**Claims And Evidence:**

Yes

**Requested Changes:**

- Address Cold-Start Scenarios:

You could try to incorporate strategies to handle cold-start users and items, such as leveraging content-based features or meta-learning techniques. You could include experiments to evaluate the method’s performance in cold-start scenarios.

- Improve Clustering Mechanisms

You could try to improve the clustering process by e.g. integrating multi-modal features (e.g., textual or visual data) or adaptive clustering techniques to ensure high-quality item categorization.

- Provide Practical Guidelines

Please include practical recommendations for deploying the method, such as resource requirements and tips for hyperparameter tuning. This is fundamental for such a complicated method. Also, if possible, explore automated tuning strategies like Bayesian optimization to reduce the complexity of selecting the many hyperparameters of the proposed approach.

- Expand Dataset Diversity

If possible, test the method on additional datasets. Useful scenarios may be the datasets with extreme sparsity, dense interactions, or highly dynamic user-item relationships. This can further validate the generalizability of the proposed approach

**Strengths And Weaknesses:**

S1: mitigates the issue of catastrophic forgetting: The proposed approach effectively addresses the problem of catastrophic forgetting in incremental learning by maintaining a personalized negative reservoir that balances stability and plasticity. This ensures the system retains past knowledge while adapting to new user interaction data.

S2: can be integrated with SOTA graph NN algorithms

S3: strong performance across different datasets: empirical results demonstrate significant improvements in metrics like Recall@20 and NDCG on five diverse datasets, showing the method enhances the performance of SOTA algorithms in many different scenarios.

S4: very interesting ablation studies: The paper includes thorough ablation studies that validate the importance of key components, such as the negative reservoir and clustering mechanisms.

---
---

W1: Increased complexity: the proposed approach increases the complexity, both computational and conceptual. It increases the computational complexity because it requires a set of additional steps, such as tracking user interest shifts, maintaining and updating a negative reservoir, and clustering items dynamically over time. These steps increase the overall complexity of training and may require additional computational resources, which may be problematic for large-scale systems with millions of users and items.

W2: no investigation of cold-start scenarios: The paper does not explicitly address how the approach handles cold-start users (new users with little interaction history) or cold-start items (new items with few interactions). Since incremental learning often struggles with cold-start issues, the lack of a dedicated strategy in this method could limit its applicability in such scenarios. It would be nice if the paper explored such scenarios.

W3: Presence of many hyperparameters: The approach introduces several hyperparameters, such as reservoir size and loss weights, which require careful tuning.

---

> ### Author Response · Authors · 2024-12-28
>
> We would like to thank the Reviewer for engaging with our work and providing insightful suggestions. Our detailed response is below. If you have any additional concerns, please reach out to us to discuss them.

---

> ### Author Response · Authors · 2024-12-28
> **Complexity**
>
> > **Complexity:** The proposed approach increases the complexity, both computational and conceptual. It increases the computational complexity because it requires a set of additional steps, such as tracking user interest shifts, maintaining and updating a negative reservoir, and clustering items dynamically over time. These steps increase the overall complexity of training and may require additional computational resources, which may be problematic for large-scale systems with millions of users and items.
>
> We appreciate the reviewer’s concern regarding the increased training complexity of the proposed approach. The primary contributor to the computational overhead is the item ranking process. As discussed in Section 6.5 (RQ3), we argue that performing item ranking a limited number of times during incremental training represents a reasonable trade-off between efficiency and performance. We demonstrate that the computational complexity of our method is comparable to existing approaches for generating hard negatives, such as PNS (Rendle & Freudenthaler, 2014) and NS Cache (Zhang et al., 2019), both of which also involve item ranking to identify hard negatives.
>
> A similar rationale applies to the additional memory cost associated with storing negative items for each user. This memory overhead is inherent in methods like NS Cache (Zhang et al., 2019) as well as in our model. However, this cost can be mitigated by offloading the negative item reservoir to RAM or disk storage, rather than retaining it in GPU memory, thereby reducing the memory burden.
>
> With regard to conceptual complexity we have improved the manuscript in the revision to make it more easy to read and provide the research code to make our method more accessible.

---

> ### Author Response · Authors · 2024-12-28
> **Cold Start**
>
> > **Cold-Start Scenarios:** You could try to incorporate strategies to handle cold-start users and items, such as leveraging content-based features or meta-learning techniques. You could include experiments to evaluate the method’s performance in cold-start scenarios.
>
> We agree with the reviewer that handling cold-start users and items is a critical challenge for improving the performance of recommender systems. However, it is important to note that cold-start is a well-known problem that is distinct from the focus of our work, which is centered on balancing plasticity and stability for **existing** users. We have clarified this distinction in the revised manuscript, where we explicitly state in the problem definition (Section 3) that addressing cold-start users is not the problem we aim to solve.

---

> ### Author Response · Authors · 2024-12-28
> **Clustering Mechanism Improvement**
>
> > **Improve Clustering Mechanisms** You could try to improve the clustering process by e.g. integrating multi-modal features (e.g., textual or visual data) or adaptive clustering techniques to ensure high-quality item categorization.
>
> We agree with the reviewer that more elaborate clustering mechanisms can be used. In fact, a core feature of our algorithm is that it does not really hinge on any specific clustering algorithm and that a wide variety of options is available, as demonstrated in the ablation study with an alternative clustering algorithm. We note that focusing too much on a specific clustering algorithm choice can detract from our overall focus and contribution which is the negative reservoir based on user interest shift. The selected clustering algorithm is not a contribution of this paper, but merely an off-the-shelf tool that we employ.

---

> ### Author Response · Authors · 2024-12-28
> **Practical Guidelines**
>
> > **Practical Guidelines** Please include practical recommendations for deploying the method, such as resource requirements and tips for hyperparameter tuning. This is fundamental for such a complicated method. Also, if possible, explore automated tuning strategies like Bayesian optimization to reduce the complexity of selecting the many hyperparameters of the proposed approach.
>
> We thank the reviewer for raising this point which we address by improving the clarity of the paper. The hyperparameters introduced by our method are outlined in Appendix H. We note that the main components of the method that require a design choice are (i) the clustering algorithm's number of clusters, (ii) the loss balancing term $\beta$ in eq. (16), and (iii) the size of the reservoir.
>
> In the revised manuscript we include a new Appendix K that explains step-by-step how to practically tune the hyperparameters on a new dataset including the data prerequisites (number of blocks), the tuning procedure to select good hyperparameters for the number of clusters, $\beta$ and the size of the reservoir.

---

> ### Author Response · Authors · 2024-12-28
> **Expand Dataset Diversity**
>
> > **Expand Dataset Diversity** If possible, test the method on additional datasets. Useful scenarios may be the datasets with extreme sparsity, dense interactions, or highly dynamic user-item relationships. This can further validate the generalizability of the proposed approach
>
> Thank you for the constructive feedback. We add results for MovieLens10M in the revision. According the results, our negative reservoir is confirmed to also improve upon the the base models (GraphSAIL, SGCT, LWC-KD). This brings the total datasets that examine our model to 6 datasets with varying degrees of sparsity.
>
> |      Model      |  Inc 1 |  Inc 2 |  Inc 3 | Average | $\Delta$\% |
> |:---------------:|:------:|:------:|:------:|:-------:|:----------:|
> |    Fine Tune    | 0.0923 | 0.0904 | 0.0957 |  0.0918 |    0.00    |
> |      LSP\_s     | 0.0912 | 0.0923 | 0.0941 |  0.0925 |    0.76    |
> |     Uniform     | 0.0874 | 0.0879 | 0.0902 |  0.0885 |    -3.59   |
> |   Inv\_degree   | 0.0906 | 0.0911 | 0.0928 |  0.0915 |    -0.32   |
> |    GraphSAIL    | 0.0867 | 0.0887 | 0.0917 |  0.0890 |    -3.05   |
> | GraphSAIL-SANE (ours) | 0.0887 | 0.0915 | 0.0961 |  0.0921 |  **0.33**  |
> |       SGCT      | 0.0894 | 0.0902 | 0.0960 |  0.0919 |    0.11    |
> |    SGCT-SANE (ours)  | 0.0927 | 0.0932 | 0.0980 |  0.0946 |  **3.05**  |
> |      LWC-KD     | 0.0902 | 0.0910 | 0.0973 |  0.0928 |    1.09    |
> |   LWC-KD-SANE (ours)  | 0.0928 | 0.0953 | 0.1012 |  0.0964 |  **5.01**  |

---

### Review · Reviewer_2utX · 2024-12-10

**Summary Of Contributions:**

The paper proposes a framework for continually learning a GNN recommendation model to quickly adapt to the changing sets of users, items, and trends in their affinity. This is an important and fundamental problem in many recommendation systems, ranging from ads, to news and video recommendation. The framework includes a non-uniform reservoir sampling technique, where the sampling distribution is based on clusters of items, either statically determined by a taxonomy, or dynamically learned by clustering.

**Audience:**

Yes

**Broader Impact Concerns:**

I do not see broader impact concerns.

**Claims And Evidence:**

No

**Requested Changes:**

1. Please move the hypothesis from page 7 to section 5.1 to to emphasize that this is a hypothesis, and not something that we know to be true
2. Separate the core contribution from the 'peripherals' more clearly. For example, the 'framework' in section 5.4 may be with or without clustering, since the taxonomy of items may be given.
3. Explain from the very beginning that the paper is about GNN recommenders with user/item identity.
4. Clarify, as explained in the weaknesses, what exactly is the contribution of this paper, and what is adoption of ideas from other papers. The distinction needs to be more clear (i.e. the T distribution).
5. Clarify how the experiments were conducted, and how the measurements were made. Otherwise, it's hard to judge the evidence provided.
6. Anonymizer and add a link to the code.

**Strengths And Weaknesses:**

# Strengths
- Solves an extremely important problem of adapting to the changing marketplace of users and items quickly and efficiently. In many practical applications, the latency between events happening, and a model trained on those events being deployed and used has a tremendous effect on the performance in practice.
- Shows significantly positive results on some recsys datasets.
- The method does not degrade the inference time in any way, which is extremely important in practical recommender systems. The cost of recommendation in many cases makes the difference between a useful and a useless method.

Beyond the strengths, it appears the paper needs some over-haul before it's ready for publication. See weaknesses and requested changes.

# Weaknesses
The weaknesses are divided into weaknesses concerning the scientific contribution and content, and weaknesses concerning the structure.

## Contribution weaknesses
1. One important paradigm of incremental learning is the paradigm of online learning - training on each example only once. It should be explained why this method is not applicable for the discussed setup of the model, or be added as a baseline.
2. The method appears to be non-extensible to the case when users or items are *not* represented by their identity, but rather only by their features (i.e. no user / item nodes at all). This needs to be explicitly written in the conclusion section, since many recsys where there isn’t a huge amount of training data and sparsity is abundant (such as ads, for example), work this way.
3. A well-known benchmark family for RecSys are the movie lens datasets. They appear to be missing.
4. A simple online-learning baseline without triplet loss is missing. It's another common method of adapting to the changing trends. See [1] for example. (i.e. binary labels, positive and negative, where the negatives are randomly sampled).
5. Lacks a link to the code (anonymized with some service, such as https://anonymous.4open.science service)

## Structural weaknesses
1. The paper appears to lack crispness and polish. It's not clear where the boundary between the core contribution and the 'peripheral' things goes, and it needs to be deduced as we read. For example:
  1.1 The paper does not explain from the very beginning, the abstract and the intro, that the main focus is GNNs for rankings. Many recommenders are not of this structure, i.e. Factorization-Machine based ad recommenders.
  1.2 In Section 5.3, it is not clear from the text weather the T distribution as the cluster assignment probability is the authors’ contribution, or the review of previous works cited at the beginning of the section. The same for the clustering method, and the square+normalization operation.
2. Section 5.2 - if the item categories are not given, how do you update the assignment of item to categories in the incremental learning setting? Based on which embeddings? How does it work if the embeddings constantly change? What about new items from the current batch for which we don't have reasonable embeddings yet? Or is the negative reservior only from the previous batches? A lot of missing details here. The code, again, would help, even if these details are not in the paper.
3. In Section 5.3 -  what does “ To obtain confident cluster assignments” mean? “Confident” in what sense?
4. The content 5.4 should probably appear before the components of the framework are described. A top-down approach is typically more readable than a bottom-up approach. Moreover, why is $beta_1 = 1$ a reasonable choice?
5.  Section 5.1 is hard to follow, and there’s no intuition on *why* we should consider several independent observations of the same negative item. The hypothesis on page 7 should come earlier, to make the paper more readable.
6. It's not clear in the experiment section how the measurements are conducted (per-block? Only last block? Something else? How the blocks are aggregated?), and how the data-sets are divided into blocks.


[1]: Aharon, M., Kagian, A., & Somekh, O. (2017, April). Adaptive online hyper-parameters tuning for ad event-prediction models. In Proceedings of the 26th International Conference on World Wide Web Companion (pp. 672-679).

---

> ### Author Response · Authors · 2024-12-28
>
> We would like to thank the reviewer for the thorough and insightful feedback. Below is our detailed response. We believe that the clarification regarding the scope of our claims, along with the additional experimental results, provides sufficient evidence to support our conclusions. Should you have any further concerns, please do not hesitate to reach out for further discussion.

---

> ### Author Response · Authors · 2024-12-28
> **Online Learning**
>
> > **Contribution 1 - Online Learning** One important paradigm of incremental learning is the paradigm of online learning - training on each example only once. It should be explained why this method is not applicable for the discussed setup of the model, or be added as a baseline.
>
> Thank you for your comment. While online learning methods can be used in our setup they are not likely to be perform well. The evidence for this comes from the first baseline we compare to. The Fine Tune baseline performs model updates from the data stream while discarding old data. This method struggles and is overall one of the weaker baselines.

---

> ### Author Response · Authors · 2024-12-28
> **Method Scope**
>
> > **Contribution 2 - Method Scope**  The method appears to be non-extensible to the case when users or items are not represented by their identity, but rather only by their features (i.e. no user / item nodes at all). This needs to be explicitly written in the conclusion section, since many recsys where there isn’t a huge amount of training data and sparsity is abundant (such as ads, for example), work this way.
>
> We agree with the reviewer that the method is not extensible to non-graph recommendation systems as well as the the case when users or items are not represented by their identity, but rather only by their features. We note that we make no claim that our method would be applicable in such a setting, in fact in our problem statement we explicitly assume the existence of a user item graph that evolves over time with updates defined in equation (1). We also state that in the second sentence of the conclusion "Our approach is easy to implement on top of any graph-based recommendation system backbone [...]".
>
> To make this point even more clear in the revised manuscript we add this sentence in the conclusion section "The method is not applicable to the case where users or items are not represented by their identity, but rather only by their features (i.e. no user / item nodes at all)".

---

> ### Author Response · Authors · 2024-12-28
> **Additional Dataset: MovieLens**
>
> **Contribution 3 - MovieLens**  A well-known benchmark family for RecSys are the movie lens datasets. They appear to be missing.
>
> Thank you for the constructive feedback. We add results for MovieLens10M in the revision. According to the results, our negative reservoir is confirmed to also improve upon the the base models (GraphSAIL, SGCT, LWC-KD). This brings the total datasets that we evaluate our model on to 6 with varying degrees of sparsity. Our proposed negative reservoir improves the base models in all MovieLens experiments.
>
> |      Model      |  Inc 1 |  Inc 2 |  Inc 3 | Average | $\Delta$\% |
> |:---------------:|:------:|:------:|:------:|:-------:|:----------:|
> |    Fine Tune    | 0.0923 | 0.0904 | 0.0957 |  0.0918 |    0.00    |
> |      LSP\_s     | 0.0912 | 0.0923 | 0.0941 |  0.0925 |    0.76    |
> |     Uniform     | 0.0874 | 0.0879 | 0.0902 |  0.0885 |    -3.59   |
> |   Inv\_degree   | 0.0906 | 0.0911 | 0.0928 |  0.0915 |    -0.32   |
> |    GraphSAIL    | 0.0867 | 0.0887 | 0.0917 |  0.0890 |    -3.05   |
> | GraphSAIL-SANE (ours) | 0.0887 | 0.0915 | 0.0961 |  0.0921 |  **0.33**  |
> |       SGCT      | 0.0894 | 0.0902 | 0.0960 |  0.0919 |    0.11    |
> |    SGCT-SANE (ours)   | 0.0927 | 0.0932 | 0.0980 |  0.0946 |  **3.05**  |
> |      LWC-KD     | 0.0902 | 0.0910 | 0.0973 |  0.0928 |    1.09    |
> |   LWC-KD-SANE (ours)  | 0.0928 | 0.0953 | 0.1012 |  0.0964 |  **5.01**  |

---

> ### Author Response · Authors · 2024-12-28
> **Baseline without Triplet Loss**
>
> > **Contribution 4 - Baseline without Triplet Loss**  A simple online-learning baseline without triplet loss is missing. It's another common method of adapting to the changing trends. See [1] for example. (i.e. binary labels, positive and negative, where the negatives are randomly sampled).
> [1]: Aharon, M., Kagian, A., & Somekh, O. (2017, April). Adaptive online hyper-parameters tuning for ad event-prediction models. In Proceedings of the 26th International Conference on World Wide Web Companion (pp. 672-679).
>
> We thank the reviewer for pointing out this work which will be added to our review section.
> For a comparison to a purely online learning based approach in our setting see our response to **Contribution 1 - Online Learning** comment.
>
> We note that direct comparison with Aharon et al. 2017 [1] is not applicable as this work focuses on CTR prediction. This inherently changes the nature of the problem as CTR measures user engagement by tracking clicks, while top-k evaluates the relevance and quality of the top recommendations. Therefore, the loss, model architecture and access to features of a CTR and a top-k model are fundamentally different and do not admit direct comparison.
>
> More broadly, we note that our proposed approach is ONLY applicable in a graph setting where triplet loss is used. Our contribution does not include CTR tasks, non-graph based models or non triplet based model training. This is mentioned in the first bullet point of our contributions in the introduction section: *"This is the first work to propose a negative reservoir design tailored for incremental learning in graph-based recommender systems. The approach, for which we provide a principled mathematical derivation in Section 5.1, is compatible with existing learning frameworks that involve triplet loss."*

---

> ### Author Response · Authors · 2024-12-28
> **Code**
>
> > **Contribution 5 - Code**  Lacks a link to the code (anonymized with some service, such as https://anonymous.4open.science service).
>
> We have included a zipped file in the supplementary with the code. We understand that this might not be the most accessible way to view / share the code so we have created an anonymous repo at (https://anonymous.4open.science/r/GraphSANE-DDCE/README.md). We commit to making our code public and providing a GitHub link in the camera ready version of the paper upon paper acceptance.

---

> ### Author Response · Authors · 2024-12-28
> **Clear Contribution**
>
> **Writing 1 - Clear Contribution** The paper appears to lack crispness and polish. It's not clear where the boundary between the core contribution and the 'peripheral' things goes, and it needs to be deduced as we read. For example: 1.1 The paper does not explain from the very beginning, the abstract and the intro, that the main focus is GNNs for rankings. Many recommenders are not of this structure, i.e. Factorization-Machine based ad recommenders. 1.2 In Section 5.3, it is not clear from the text weather the T distribution as the cluster assignment probability is the authors’ contribution, or the review of previous works cited at the beginning of the section. The same for the clustering method, and the square+normalization operation.
>
> Our main contributions are outlined in the final paragraph of the introduction (Section 1). Specifically, we make the following claims:
>
> - We introduce a negative reservoir design tailored for incremental learning in graph-based recommender systems using triplet loss.
> - We demonstrate that incorporating this reservoir into established graph incremental frameworks results in significant and consistent improvements over recent state-of-the-art (SOTA) incremental learning techniques.
> - We further investigate that these improvements stem from the fact that prior works assume static user interests when selecting negative items. In Section 6.6, we show that our model can rapidly adapt to users with shifting interests, leading to enhanced recommendations. Thus, our framework fosters better adaptability for users whose interests evolve over time.
>
> We strongly believe that our experiments substantiate these three contributions.
>
> We agree with the reviewer that the lack of explicit mention of graph recommender systems in the abstract could reduce the clarity of the topic. In the revised manuscript, we have added a sentence to the abstract specifying that our technique applies to graph recommender systems. Although the introduction already highlights that our work focuses on incremental learning in graph-based recommender systems (as noted in the first bullet point of the contributions subsection), we have also included a clearer statement about the scope earlier in the section for better clarity in the revised manuscript.
>
> Regarding Section 5.3, we do not claim any novel contributions here, as our paper does not list any clustering algorithm as a contribution in the introduction. As discussed at the beginning of Section 5.3, "the deep structural clustering method we use is adopted from two recent works (Bo et al., 2020) and (Wang et al., 2019)." This section merely summarizes existing end-to-end trainable clustering methods that we adopt. We elaborate on this algorithm to emphasize that, while our method requires a clustering technique, it can be trained end-to-end without the need for separate training procedures for clustering and model weight updates. In the revision, we make it clearer that Section 5.3 discusses an existing clustering algorithm without leaving room for interpretating this section as a novel contribution.
>
> In summary, the following revisions have been made in the updated version of the paper:
> - The abstract has been edited to explicitly acknowledge that our contributions are within the scope of incremental learning for graph-based recommender systems.
> - The scope of our work has been reiterated earlier in the introduction.
> - Non-graph-based online learning works are acknowledged in the literature review section, with clarification that they fall outside the scope of our contributions.
> - Section 5.3 now clearly states that the clustering algorithm we use is not a research contribution, but simply an adopted method.
>
> The relevant changes are highlighted in green in the revised manuscript.

---

> ### Author Response · Authors · 2024-12-28
> **Clarity**
>
> > **Writing 2 - Section 5.2 Clarity**
>
> > If the item categories are not given, how do you update the assignment of item to categories in the incremental learning setting? Based on which embeddings?
>
> In general, we assume that item categories are not given. We use a clustering technique to assign item embeddings (from the GNN representation of each item node) to clusters and interpret each cluster as a pseudo-category. Specifically, as outlined in Section 5.3 we update the assignment of item categories using soft cluster memberships. Every training epoch we model the probability of an item belonging to a cluster using the soft membership in equation (15) based on item embedding location in latent space (GNN embeddings of item nodes of the user-item graph).
>
> > How does it work if the embeddings constantly change?
>
> We note that we jointly train the GNN model weights that produce the embeddings as well as the clustering algorithm in $\mathcal{L}_{\text{TOTAL}}$ equation (16)). Therefore, the cluster assignments and GNN weights that derive the embeddings are always in sync, so we can account for this change dynamically.
>
> > What about new items from the current batch for which we don't have reasonable embeddings yet? Or is the negative reservior only from the previous batches? A lot of missing details here.
>
> Our negative reservoir tracks shifts in user interest over time; however, we do not apply this approach to new users, as there is no historical interaction data available to observe such shifts. We consider the cold-start problem to be distinct from our setting. We note that, since our framework assumes small and frequent updates for each data block, the proportion of new users and items within each incremental training block is generally small.
>
> We note that, since our framework (and all other baseline models we compare to) assume small and frequent updates for each data block, the proportion of new users and items within each incremental training block is generally small. The cold user and item representations are learned via the standard backbone and improving performance for them is beyond our scope.
>
> In the revised manuscript, we explicitly clarify that our approach does not address cold-start users or items in the  problem statement (Section 3).

---

> ### Author Response · Authors · 2024-12-28
> **Question**
>
> > **Writing 3 - Question** In Section 5.3 - what does “ To obtain confident cluster assignments” mean? “Confident” in what sense?
>
> The clustering algorithm we use provides soft cluster memberships. The cluster memberships are encoded via a vector that, for a given item, contains the probability of said item belonging to each of the clusters. As explained by Bo et al. (2020) and Wang et al. (2019), an issue with equation (12) is that it can sometimes provide diffuse cluster assignments (close to uniform). Broadly speaking "confident" in our setting can refer to low entropy cluster probability vector where the majority of the probability mass is concentrated in one cluster. Thus to encourage more confident assignments, i.e., more probability mass on a single cluster, it is transformed via a square normalization to obtain equation (13).
>
> In the revised manuscript we include sentences from the above paragraph to make Section 5.3 more clear.

---

> ### Author Response · Authors · 2024-12-28
> **Rearrange Content**
>
> > **Writing 4a - Rearrange Content** The content 5.4 should probably appear before the components of the framework are described. A top-down approach is typically more readable than a bottom-up approach.
>
> Thank you for your suggestion. We recognize that a top-down approach may be more accessible to some readers, while others may find a bottom-up approach easier to follow. Some readers may prefer to first understand the overall framework before delving into the specifics of each component (top-down), while others may find it overwhelming to be introduced to all components in Section 5.4 before discussing them in detail in Sections 5.1, 5.2, and 5.3.
>
> To accommodate both types of readers, we revise the manuscript by moving part of the content from Section 5.4 to the beginning of Section 5 to provide a high-level overview of the approach. This allows readers who prefer a quick summary to understand the core components of our method early on, without being overwhelmed by excessive detail before exploring integration details of the components further in Section 5.4.

---

> ### Author Response · Authors · 2024-12-28
> **Question**
>
> > **Writing 4b - Question** Moreover, why is $\beta_1=1$ a reasonable choice?
>
> Since both \mathcal{L}_{\text{BASE}} and \mathcal{L}_{\text{SANE}} consist of triplet loss terms, \mathcal{L}_{\text{SANE}} is of the same scale as \mathcal{L}_{\text{BASE}}. Therefore, we set $\beta_1 = 1$. It is important to note that \mathcal{L}_{\text{BASE}} is the sum of \mathcal{L}_{\text{TRIPLET}} and \mathcal{L}_{\text{KD}}, with the latter being much smaller in magnitude than \mathcal{L}_{\text{TRIPLET}}. In contrast, the clustering loss operates on a completely different scale and requires tuning, so $\beta_2$ varies by dataset.
>
> To clarify that $\beta_1$ can always be set to 1, we revise the total loss formulation as follows:
>
> $\mathcal{L}_{\text{TOTAL}} = \mathcal{L}_{\text{BASE}} + \mathcal{L}_{\text{SANE}} + \beta \mathcal{L}_{\text{KL}}$,
>
> where we have removed $\beta_1$ since it is always set to 1, to avoid potential confusion about it being a tunable hyperparameter. Additionally, we replace $\beta_2$ with a single parameter, $\beta$.
>
> These changes have been added to Section 5.4 of the manuscript.

---

> ### Author Response · Authors · 2024-12-28
> **Move hypothesis earlier**
>
> > **Writing 5 - Move hypothesis from page 7 earlier** Section 5.1 is hard to follow, and there’s no intuition on why we should consider several independent observations of the same negative item. The hypothesis on page 7 should come earlier, to make the paper more readable.
>
> We thank the reviewer for this recommendation that can help improve the readability of the manuscript. We note that have introduced the hypothesis (at a higher level of abstraction) in the second paragraph of page 2 near the end of the introduction section: "We identify two unique challenges for designing a good negative sampler for an incremental learning framework [...]".
>
> We agree that this sentence is somewhat vague when it comes to exact methodological hypothesis so in the revision we include more details in the final paragraph of the introduction with our explicit hypothesis to better motivate the methodology before we present it.

---

> ### Author Response · Authors · 2024-12-28
> **Dataset Setup**
>
> > **Writing 6 - Dataset Setup** It's not clear in the experiment section how the measurements are conducted (per-block? Only last block? Something else? How the blocks are aggregated?), and how the data-sets are divided into blocks.
>
>
> We thank the reviewer for highlighting the lack of clarity in the data pre-processing methodology. In this work, we adopt the data pre-processing approach described in the literature. Our data setup is identical to the baselines of GraphSAIL (Xu et al., 2020), Inverse Degree (Ahrabian et al., 2021), SGCT, and LWC-KD (Wang et al., 2021) — refer to the full bibliographic citations in the paper.
>
> For all datasets, we chronologically sort the user-item interactions and use base blocks containing 60% of the data to pre-train an initial GNN model backbone, which is shared across all incremental techniques. Subsequently, for each incremental block, we use 10% of the data, resulting in four incremental blocks. At each stage of incremental learning, we train on the current block and test the algorithm on the following block. The final incremental block is exclusively used for testing, and no training is performed on it. For evaluation, we record the performance for each incremental block and report the average.
>
> It is important to note that, because the data blocks are ordered temporally, there is no leakage of future information during training. To further clarify the data pre-processing steps, we have added a new Appendix J, which includes Fig. 5, in the revised manuscript.

---

> ### Comment · Reviewer_2utX · 2024-12-28
> **Response**
>
> Thank you for the revision and the comments. Most of my concerns have been resolved. Only one has not been, but it's minor - CTR prediction is not only a task, but a tool. Predicted CTR can also be used fir ranking. In fact, its one of tbe "pointwise losses" in the "unbiased learning to rank" literature.
>
> However, i believe that methods dedicated directly to ranking outperform point wise losses.

---

> > ### Author Response · Authors · 2024-12-29
> >
> > We are glad to see that most of the reviewers concerns have been resolved.
> >
> > Regarding the CTR prediction, we agree with the reviewer that typically methods dedicated directly to ranking outperform point wise losses.
> >
> > In the camera ready version of the manuscript we will include a statement to justify the baseline selection and explain why only rank-based methods were chosen.

---

### Author Response · Authors · 2024-12-28
**Revision Summary**

We are pleased that the Action Editor and the Reviewers appreciate our manuscript. We have addressed the points raised in the reviews. We hope that you find the current version ready to be published.

In response to the feedback we have:
- (Reveiewer 2utX): Clarified that the scope of our contributions is within the area of incremental learning for graph-based recommender systems.
- (Reviewers 2utX, r8Jp): Strengthened the experimental section by adding new experiments on MovieLens10M. With the addition of the new dataset we have now evaluated the nmodel on 6 total datasets, yielding good results in all of them.
- (Reviewer UQg6): Added two more negative sampler baselines from the Knowledge Graph literature (DeMix, ESNS) to further support the need for our method in the graph recommendation incremental learning setting.
- (Reviewer r8Jp): Added practical advice on how to deploy the method.

For convenience we used software to highlight the paper changes in GREEN. See updated manuscript: https://openreview.net/pdf?id=jrUUk5Fskm

Sincerely yours,
The Authors

---

### Decision · Action_Editor_iGxv · 2025-01-20

**Recommendation:** Accept with minor revision

**Comment:**

After the discussion, the reviewers were unanimous that this paper should be accepted. Your responses and updated manuscript answered all of the reviewers' questions and comments.

As you had promised in your response to reviewer UQg6, please add a discussion "indicating that the focus of our work is on accounting for user interest shifts rather addressing the problem from a traditional continual learning perspective that focuses on preventing catastrophic forgetting in the final version of the manuscript." That's the only requested revision.

Congratulations!

**Audience:**

This paper will likely be interseting both to the recsys audience and to the continual learning audience (especially those looking for current real-world applications).

**Claims And Evidence:**

After the discussion period, the reviewers unanimously stated that the claims were correctly supported.

---

> ### Author Response · Authors · 2025-02-11
>
> Dear AE,
>
> We have uploaded the camera-ready version of the paper.
>
> The paper includes the clarification reviewer UQg6 requested in paragraph 3 of the introduction section. The paragraph now states the following: "[...] We note that the focus of our work is on accounting for user interest shifts rather addressing the problem from a traditional continual learning perspective that focuses on preventing catastrophic forgetting (Wang et al., 2023). The reason for this is that, as we show in the experiments and case study, existing incremental learning techniques are too focused on stability and neglect model plasticity."
>
> We have also added public links to the GitHub repository containing our method's implementation and experiment code in the experiments section.
>
> Thank you very much for overseeing our paper!